

# Assessing the sensitivity of the Vanderford Glacier, East Antarctica, to basal melt and calving

Lawrence A. Bird[1], Felicity S. McCormack[1], Johanna Beckmann[1], Richard S. Jones[1], and Andrew N. Mackintosh[1]

[1]Securing Antarctica's Environmental Future, School of Earth, Atmosphere and Environment, Monash University, Clayton, Kulin Nations, Victoria, Australia

**Correspondence:** Lawrence A. Bird (lawrence.bird@monash.edu)

## Abstract

Vanderford Glacier is the fastest retreating glacier in East Antarctica; however, the dominant driver of the observed grounding line retreat remains largely unknown. The presence of warm modified Circumpolar Deep Water offshore Vanderford Glacier suggests that grounding line retreat may be driven by ice shelf basal melt, similar to the

neighbouring Totten Glacier. Here, we use an ice sheet model to assess the relative contributions of basal melt and calving to mass loss and grounding line retreat at Vanderford Glacier. We compare simulations forced both by satellite-derived estimates of basal melt and calving, and varying magnitude idealised basal melt and ice-front retreat. Observed basal melt rates are too low to drive grounding line migration; instead, basal melt rates in excess of 50 m yr$^{-1}$ at the grounding line are required to generate grounding line retreat similar to observations. By contrast,

calving experiments suggest that $> 80$ % ice-front retreat - well in excess of the observed ice-front retreat since 1996 - needs to occur to generate grounding line retreat similar to observations. Our results suggest that grounding line retreat and dynamic mass loss at Vanderford Glacier is likely to be dominated by basal melt, with an almost negligible contribution from calving. However, basal melt rates that generate grounding line retreat in our idealised experiments are twice the current estimates, highlighting the need for improved constraints on basal melting in the

Vincennes Bay region.

## 1   Introduction

Vanderford Glacier (Fig. 1) is the dominant outlet glacier of the Vincennes Bay drainage basin, which contains approximately 0.67 m of global sea level equivalent (Morlighem et al., 2020) and is a sub-basin of the Aurora Subglacial Basin, East Antarctica. Vanderford Glacier has experienced approximately 18.6 km of grounding line retreat between

1996 and 2020 (~0.8 km yr$^{-1}$) and is the fastest retreating glacier in East Antarctica, dwarfing equivalent rates of 0.2 km yr$^{-1}$ for neighbouring glaciers within the Vincennes Bay drainage basin (Picton et al., 2023; Stokes et al.,





2022). Vanderford Glacier retreat also exceeds that of nearby major outlet glaciers of the Aurora Subglacial Basin, Totten and Moscow University, which have retreated $3.51 \pm 0.49$ km and $13.85 \pm 0.08$ km, respectively, over the same period (Li et al., 2023b). The large spread in grounding line retreat rates between the Vincennes Bay glaciers

(Picton et al., 2023) implies that localised conditions likely control Vanderford Glacier retreat.

Neighbouring the Vincennes Bay drainage basin, the Totten Glacier drainage basin is the largest catchment of the Aurora Subglacial Basin (Fig. 1). Totten Glacier has been losing mass at an increasing rate over recent decades and represents the largest East Antarctic contributor to sea-level rise (Li et al., 2023a). McCormack et al. (2023) demonstrate the close relationship between Totten and Vanderford Glaciers' flow configurations, suggesting that

small changes to local ice geometry (e.g. potentially associated with changes in ice-shelf buttressing at Vanderford Glacier due to ocean-driven thinning or calving) could result in large-scale reconfiguration of ice flow and subglacial hydrology, having larger implications on ice dynamics across the Aurora Subglacial Basin. Vanderford Glacier appears to be currently grounded on a prograde bed slope limiting its retreat (Fig. 2c); however, bed topography in the region upstream of the grounding line is associated with $> 500$ m of uncertainty, which is some of the highest in East

Antarctica (Morlighem et al., 2020). The region may be susceptible to Marine Ice Sheet Instability (MISI; Schoof, 2007; Weertman, 1974) under future warming conditions if extensive grounding line retreat into the upstream Aurora Subglacial Basin should occur.

Recent mass loss and grounding line retreat at Vanderford Glacier is believed to be linked to the intrusion of warm modified Circumpolar Deep Water (mCDW) into the ice-shelf cavity via a deep marine trough offshore; however,

high-resolution mapping of the trough extent across the continental shelf has not been conducted (Picton et al., 2023; McCormack et al., 2023). Basal melting in the region is likely dominated by mode 2 melting (Jacobs et al., 1992), where warm mCDW at depth drives melt close to the grounding line. This is consistent with the presence of the warmest recorded intrusions of mCDW in East Antarctica which have been observed in Vincennes Bay (Ribeiro et al., 2021; Herraiz-Borreguero and Naveira Garabato, 2022). The mCDW temperatures in Vincennes Bay are comparable

to those at Totten Glacier, where mode 2 melting is dominant (Silvano et al., 2017, 2018; Ribeiro et al., 2021). However, recent satellite-derived basal melt estimates from Paolo et al. (2022) and Davison et al. (2023) show basal melt rates ranging from -9 to 34 m yr$^{-1}$ across the Vincennes Bay ice shelves (Fig. 2a-b), compared to estimates of -8 to 67 m yr$^{-1}$ at Totten Glacier; up to twice those of Vincennes Bay, with high melt rates close to the grounding line (Fig. 2a-b). These disparate satellite-derived basal melt observations in neighbouring systems with similar offshore

ocean properties suggest that basal melt rates in Vincennes Bay may be underestimated.

Recent work has also explored the influence of basal melt and calving on both the ice-shelf mass budget (Davison et al., 2023), and changes in grounded ice discharge (Greene et al., 2022) for the Vincennes Bay region. Greene et al. (2022) model the instantaneous response to changes in floating ice geometry (calving and thinning) and show that changes in the ice-front position accounts for approximately half of all Antarctic mass loss since 2007. For

the Vincennes Bay region, Davison et al. (2023) show that calving dominated the freshwater fluxes (i.e. ice loss)



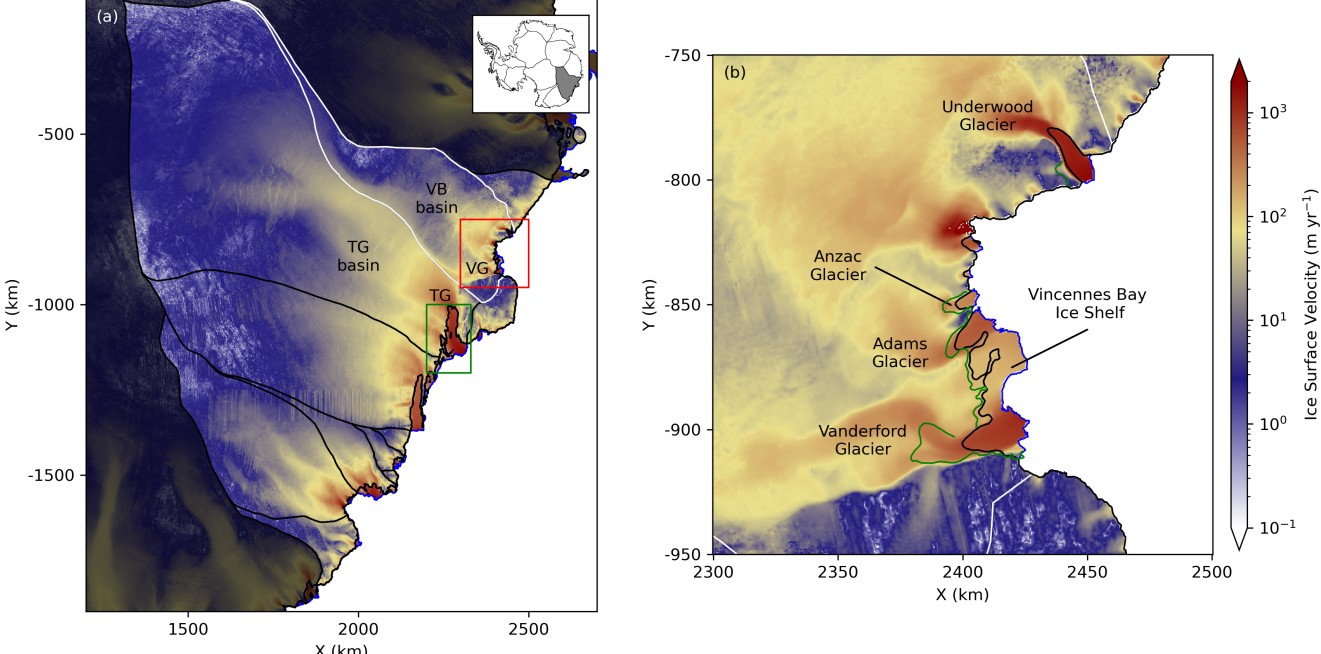

**Figure 1.** Overview of the Aurora Subglacial Basin, showing the Vincennes Bay drainage basin. (a) MEaSUREs v2 ice surface velocity (Rignot et al., 2017) for the Aurora Subglacial Basin. Black lines denote drainage divides and the MEaSUREs v2 grounding line from Mouginot et al. (2017). The model domain and Vincennes Bay drainage boundary is shown in white. Abbreviations are defined as follows: VG = Vanderford Glacier; TG = Totten Glacier; VB Basin = Vincennes Bay drainage basin; and TG Basin = Totten Glacier drainage basin. Red box indicates the extent shown in (b). The green box indicates the Totten Glacier extent shown in Fig. 2. (b) Vincennes Bay glaciers and ice shelves; all graphical components are consistent with (a). The 2020 grounding line is shown in green (Picton et al., 2023). The colour scale is consistent across (a) and (b).

from the ice shelf for the 1997-2021 period. However, to attribute relative contributions of basal melt and calving processes on mass loss relies on accurate predictions of basal melt. With no direct measurements of ice shelf melt in Vincennes Bay, basal melt estimates are derived from satellite altimetry-based methods, which require various simplifying assumptions that could lead to biases, particularly in small systems such as Vincennes Bay (Chartrand

and Howat, 2023).

The sensitivity of Vanderford Glacier grounding line retreat and ice mass loss to different magnitude basal melt and calving perturbations has not been investigated, presenting a significant gap in our understanding of the region. Given the potential for broader implications across the Aurora Subglacial Basin arising from changes at Vanderford Glacier (McCormack et al., 2023), quantifying the relative contributions of these processes on grounding line retreat

and mass loss is essential.





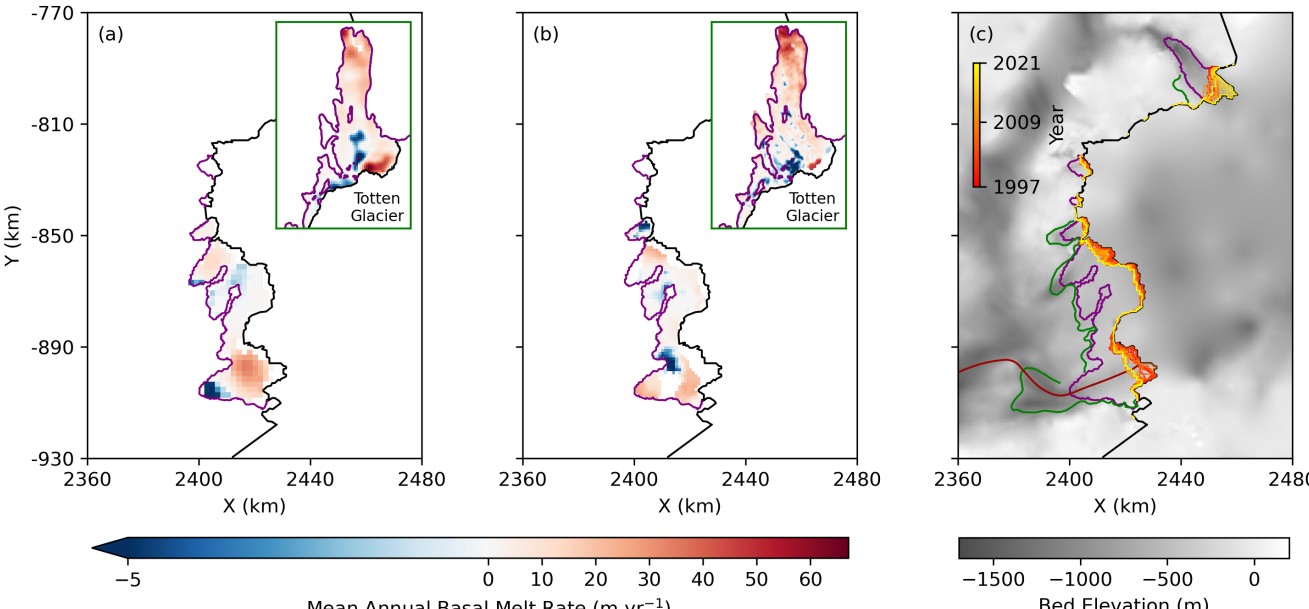

**Figure 2.** Current mass loss estimates for Vincennes Bay. (a) Mean annual basal melt rate derived from satellite altimetry (Paolo et al., 2022). (b) Mean annual basal melt rate derived from satellite altimetry (Davison et al., 2023). (c) Ice-shelf front positions from Greene et al. (2022), 2020 grounding line position (Green line) from (Picton et al., 2023), and Vanderford Glacier flow line along which grounding line retreat is measured (Red line). Bed elevation is taken from BedMachine V3 (Morlighem et al., 2020). On all panels, the model domain extent is shown in black and the initial MEaSUREs v2 grounding line (Mouginot et al., 2017) is shown in purple. The Totten Glacier inset in (a) and (b) is not to scale and the extent is shown in Fig. 1a.

The aim of this study is to determine the relative contributions of sub-ice shelf basal melt and calving to mass loss and grounding line retreat at Vanderford Glacier. Specifically, we use time-evolving numerical ice sheet model simulations to address the questions: 1) what are the relative contributions of basal melt and calving to grounding line retreat at Vanderford Glacier; and 2) can current estimates of basal melt and ice-front retreat explain recent observed grounding line retreat at Vanderford Glacier?

This paper is structured as follows: in Sect. 2, we describe the ice-sheet model set-up and initialisation process, and provide an overview of our perturbation experiments. In Sect. 3, we present the results of our ice-sheet model simulations, focusing on changes to key measures of mass loss and grounding line retreat; these results are discussed in Sect. 4. Finally, we provide a conclusion of this work in Sect. 5.



## 2 Data and methods

### 2.1 Ice sheet model setup

We use the Ice-sheet and Sea-level System Model (ISSM; Larour et al., 2012) to run transient simulations of the Vincennes Bay drainage basin (Fig. 1). The model domain covers the Vincennes Bay drainage basin from the MEaSUREs Antarctic Boundaries for International Polar Year (IPY) 2007-2009 v2 (Mouginot et al., 2017), extended to include the cumulative maximum ice-shelf front extent from 1997 to 2021 from Greene et al. (2022). We use an anisotropic mesh comprising 95,156 elements with a variable resolution ranging from ~150 m around the ice-front of the Vincennes Bay ice-shelf, to a maximum resolution of ~12 km in the interior of the basin, with a mean resolution of ~590 m across the Vincennes Bay ice shelf and upstream of the grounding line. The initial grounding line is taken from the MEaSUREs Antarctic Boundaries for IPY 2007-2009 v2 (Mouginot et al., 2017). Bed topography and ice geometry are taken from BedMachine Antarctica v3 (Morlighem et al., 2020), and surface ice velocities are from MEaSUREs v2 (Fig. 1). We assume floating ice is in hydrostatic equilibrium. Due to the paucity of bathymetry estimates below floating ice, we adjust and smooth the bathymetry in the ice-shelf cavity such that it deepens from the grounding line to the ice-front, better reflecting the deep bathymetry offshore the Vanderford ice-front. We enforce a minimum water depth of 50 m below the ice-shelf base, and lower the bathymetry at the ice-shelf front by 840 m which is the mean difference between the BedMachine bathymetry and multibeam swath bathymetry at Vanderford Glacier ice-front (Commonwealth of Australia, 2022). This approach may limit any grounding line advance across model simulations; however, the initial grounding line represents the most advanced grounding line since observations began, thus, we do not expect grounding line advance beyond this position. Furthermore, as the cavity geometry is not an explicit parameter in the basal melt parameterisation used here, deepening the bathymetry in the ice-shelf cavity does not impact melt rates. For all model simulations, we use the two-dimensional Shelfy-Stream Approximation (SSA) to the Stokes equations (MacAyeal, 1989).

Recognising the importance of the choice of basal friction law on ice mass loss estimates and grounding line dynamics (Brondex et al., 2017, 2019; Barnes and Gudmundsson, 2022), we consider different friction laws to account for the sensitivity of ice sheet models to different basal friction representations: Weertman (1974), Budd et al. (1979), and Schoof (2005), given respectively by:

$$\tau_b = C_w^2 |u_b|^{m-1} u_b \tag{1}$$

$$\tau_b = C_b^2 N u_b^{3m} \tag{2}$$



$$\tau_b = \frac{C_s^2 |u_b|^{m-1}}{(1 + (C_s^2/(C_{max}N))^{1/m}|u_b|)^m} u_b \tag{3}$$

where $\tau_b$ is basal shear stress, $C_w$, $C_b$, and $C_s$ are friction coefficients for the Weertman, Budd, and Schoof friction laws, respectively, $u_b$ is the basal velocity, $m$ is a positive exponent set to $1/3$, and $C_{max}$ is Iken's bound, set to 0.5 (Brondex et al., 2017). In Eq. 2 and Eq. 3, $N$ relates to the effective pressure, calculated following Brondex et al. (2017), assuming perfect hydrological connection between the subglacial hydrology system and the ocean, such that:

$$N = \begin{cases} \rho_i gH + \rho_w gz_b & \text{if } z_b < 0 \\ \rho_i gH & \text{if } z_b \geq 0 \end{cases} \tag{4}$$

where $\rho_i$ is the ice density (917 kg m$^{-3}$), $g$ is the gravitational acceleration (9.81 m s$^{-2}$), $H$ is the ice thickness (m), $\rho_w$ is the seawater density (1023 kg m$^{-3}$), and $z_b$ is the ice base elevation (m). We note that the system response to different friction laws is not the focus of this study; rather, we consider multiple friction laws to ensure that our findings are not dependent on a given representation of basal friction. We use a Glen-type flow relation (Glen, 1953) with a stress exponent of n = 3 and compute ice rigidity using inverse methods (Morlighem et al., 2013, Sect. 2.2). For all transient simulations, we use a 1-month time step, with sub-element grounding line (SEP1; Seroussi et al., 2014) and basal melt (SEM1; Seroussi and Morlighem, 2018) parameterisations. We provide constant surface mass balance forcing to the model using the 1991-2020 climate normal mean annual surface mass balance from RACMO2.3p2 reanalysis (van Wessem et al., 2018).

We limit our analysis to the region of the model domain which contributes to the Vincennes Bay and Anzac Glacier ice-shelves (Fig. 1), principally for the purpose that this region is the primary focus of this study and these individual ice shelves become merged across all simulations. We quantify changes in grounding line position by measuring its migration along a central flowline in the main trunk of Vanderford Glacier (Fig. 2c). The Underwood Glacier to the west (Fig. 1) is outside of our area of interest and shows a relatively persistent grounding line location throughout all simulations. For this reason, we do not consider the response of Underwood Glacier in this study.

## 2.2 Ice sheet model initialisation

We invert ice rigidity and basal friction coefficient for each friction law independently (Fig. 3), minimising a cost function which includes terms for the linear and logarithmic misfit between simulated and observed surface ice velocities (Morlighem et al., 2013). We first invert for ice rigidity over floating ice, selecting linear and logarithmic cost function coefficients such that the contributions to the total cost function are similar. For the basal friction inversion using the Budd friction law, we choose coefficients such that their contributions to the total cost function have the same order of magnitude (e.g. Morlighem et al., 2013). For the Schoof and Weertman friction laws, we choose cost





function coefficients that yield similar area-weighted RMSE velocity misfits to the Budd friction law, while minimising

the velocity misfit across the domain. Finally, we invert for ice rigidity across the whole domain using consistent cost

function coefficients across all friction laws such that the linear and logarithmic cost function contributions have the

same order of magnitude. To penalise sharp gradients in the resultant fields, we add Tikhonov regularisation to all

inversions, using L-curve analysis (Hansen, 2000) to determine the optimal regularisation parameters. Following the

inversion process, the area-weighted RMSE velocity misfits for the Budd, Schoof, and Weertman friction laws are 8.2

m yr$^{-1}$, 9.9 m yr$^{-1}$, and 8.4 m yr$^{-1}$, respectively. All spatially-varying basal friction and ice rigidity fields are held

constant for the duration of all simulations.

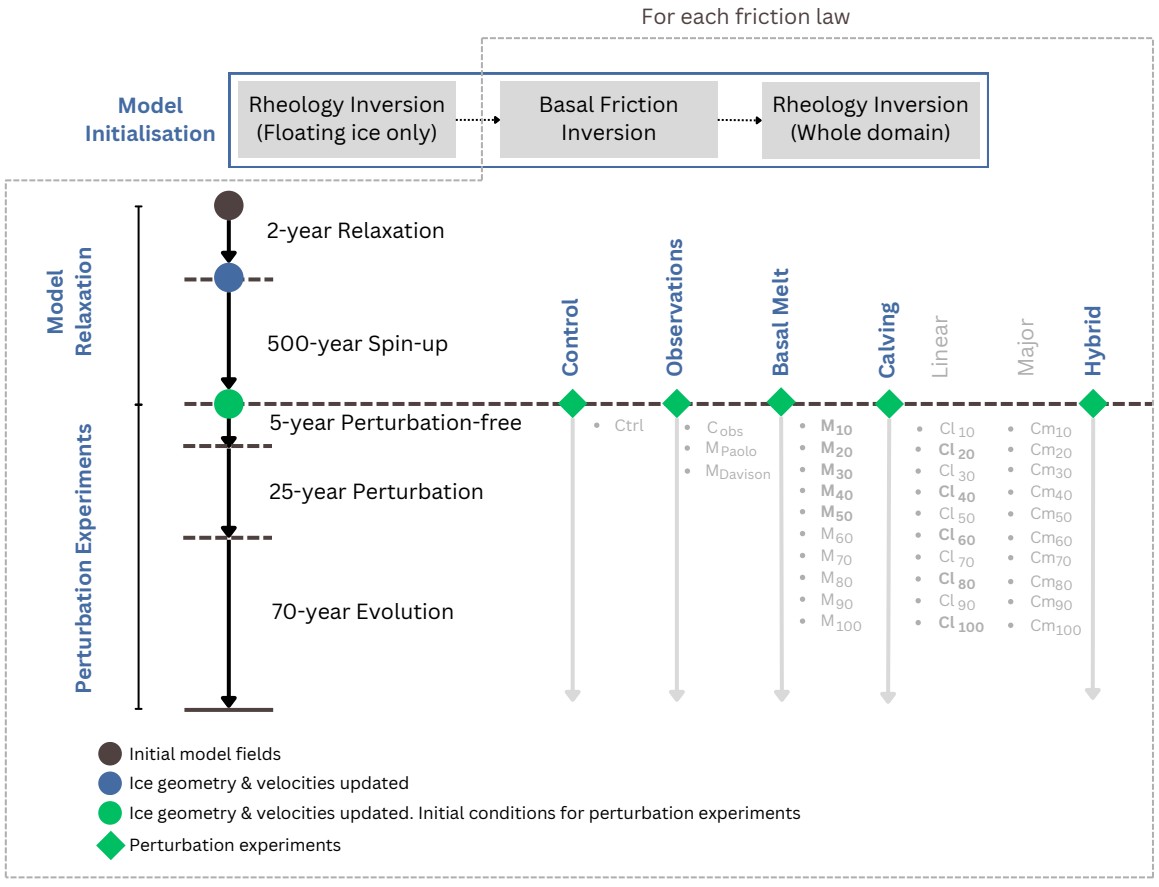

**Figure 3.** Overview of study methodology. Steps contained within the grey dotted boundary are completed independently for each friction law. Experiment names in **bold** indicate basal melt and calving processes combined in hybrid experiments.

Following model initialisation, we hold the grounding line fixed and perform an initial 2-year relaxation (Fig. 3) for

each friction law to remove non-physical artefacts that result from observational uncertainties and discrepancies

between data product epochs. Subsequently, we perform a 500-year spin-up simulation (Fig. 3) where the grounding



line is allowed to freely-evolve. For this simulation, we prescribe a baseline basal melt rate ($M$) of 3 m yr$^{-1}$ across
floating ice. We use 3 m yr$^{-1}$ as it is the median value of the two-dimensional mean annual melt rate field across
Vincennes Bay ice shelves from Paolo et al. (2022). At the end of this 500-year simulation, ice velocities and geometry
are in pseudo-steady state, and these fields are used as the initial conditions for all perturbation experiments
(Sect. 2.3).

### 2.3   Perturbation experiments

To assess the sensitivity of Vanderford Glacier to sub-ice shelf basal melt and calving, we simulate a series of
perturbation experiments. All experiments are run for a period of 100 years and for each basal friction law (Sect. 2.1).
After an initial perturbation-free 5-year period using the same constant basal melt rates and SMB as in the spin-up,
perturbations to either the basal melt rates or calving fronts are applied for 25 years, consistent with the time
period over which recent grounding line retreat has been observed at Vanderford Glacier. At the end of the 25-year
perturbation period, the model is allowed to evolve for an additional 70 years, again using the same constant forcings
applied during the spin-up simulation. This approach allows us to quantify the instant and evolving ice sheet response
to a perturbation. We also simulate a 100-year perturbation-free control simulation ($Ctrl$) where the forcings are
consistent with the spin-up simulation. The system response to the control experiment is subsequently removed from
all perturbation experiments to isolate the effect of each perturbation.

Beyond investigating the system response to observed conditions, we include basal melt and calving perturbation
outside the observed range. These perturbation experiments are intended to capture the full range of plausible
conditions, allowing us to assess the relative influence of different magnitude perturbations of different mass loss
drivers (i.e. basal melt and calving) on Vanderford Glacier. Perturbation experiments are broadly separated into four
categories as follows and are summarised in Fig. 4 and Table 1:

− *Observed mass loss* experiments assess the response of the system to observed basal melt and calving ($C_{obs}$).
          To simulate the impact of observed basal melt, we use mean annual basal melt rates derived from satellite
          altimetry for the period 1992-2017 from Paolo et al. (2022) ($M_{Paolo}$; Fig. 2a) and for the period 1997-2021 from
          Davison et al. (2023) ($M_{Davison}$; Fig. 2b). To simulate the impact of observed changes in the ice-front, we use a
          time series of approximately annual ice-front positions for the Vincennes Bay ice shelves between 1997-2021
from Greene et al. (2022) (Fig. 2c). For $M_{Paolo}$ and $M_{Davison}$ experiments, we hold the ice-front fixed and hold
          mean annual melt rates constant across the perturbation period. For $C_{obs}$ we include the baseline basal melt
          rate of 3 m yr$^{-1}$, consistent with the control simulation, and vary the ice-front position using ice masks from
          Greene et al. (2022). We apply calving events annually on January 1. There are only 24 ice masks included in
          the Greene et al. (2022) dataset, so we apply these annually from the beginning of the perturbation period
and hold the ice-front fixed following the final calving event. We note that the initial ice-front extent used in
          $C_{obs}$ simulations is the cumulative maximum ice-front extent (consistent with the control simulation), thus, the



magnitude of the initial calving event to reach the initial ice-front extent from Greene et al. (2022) is likely larger than observed events.

– *Basal melt perturbation* experiments vary the magnitude of basal melt rates applied using a simple linear parameterisation, increasing melt rate with depth, such that:

$$M = \begin{cases} M_s & \text{if } z_b \geq -300 \\ M_d & \text{if } z_b \leq -600 \end{cases} \tag{5}$$

where $M_s$ is set to 8 m yr$^{-1}$ across all experiments and represents the area-weighted mean annual melt rate from Paolo et al. (2022) and Davison et al. (2023) across regions with ice drafts $\geq$ -300 m. $M_d$ varies between experiments by increments of 10 m yr$^{-1}$, from 10 - 100 m yr$^{-1}$ and the ice-front is held fixed. Shallow and deep ice draft values (Eq. 5) are selected based on the depths of mCDW recorded within Vincennes Bay (Ribeiro et al., 2021). Basal melt rates applied to our simulations are shown in Fig. 4a.

– *Calving* experiments adjust the ice-front to simulate calving processes. Two different mechanisms of ice-front retreat (IFR) are tested, including continual linear retreat ($Cl$), intended to simulate edge-wasting, and larger calving events occurring on a 12.5-year cycle ($Cm$) such that three large-scale calving events occur within the perturbation period. The choice of a 12.5-year calving cycle is arbitrary, but is intended to represent larger discrete calving events. The final extent of IFR is consistent for $Cl$ and $Cm$ experiments, and is determined as a percentage of the distance from the mean grounding line across all friction laws to the initial ice-front in the along-flow direction. We vary the magnitude of IFR by increments of 10 %, from 10 - 100 %. Baseline basal melt is included, consistent with the control simulation. We hold the ice-front fixed at the final location following the perturbation period. Final ice-front positions for each calving perturbation are shown in Fig. 4b.

– *Hybrid* experiments combine basal melt and calving processes, described above. We explore the combined effects of basal melt and calving to determine whether comparable system responses can be generated through multiple experimental set-ups. Here, we test a subset of basal melt and calving perturbations, including IFR of 20 - 100% in increments of 20%, and $M_d$ rates of 10 - 50 m yr$^{-1}$ in 10 m yr$^{-1}$ increments. We complete these hybrid experiments using only the linear calving mechanism ($Cl$). We compare the results against the corresponding basal melt perturbation without IFR to directly assess the effect of including IFR. We assess changes only at the end of the perturbation period.

For $C_{obs}$, $Cl$, and $Cm$ experiments, we use the levelset method (Bondzio et al., 2016) to explicitly define ice-front positions for each time step, based on observations for $C_{obs}$ (Fig. 2) and idealised ice-front positions for $Cl$ and $Cm$ (Fig. 4b). The levelset method involves providing a 2D field for each time step, defining regions of ice presence (negative values) and absence (positive values), where the zero contour defines the ice-front. We assess the relative change in the perturbation experiments compared to the control experiment ($Ctrl$) for the following variables: grounding line flux ($GL_{flux}$), grounding line position, ice volume, and volume above floatation ($VAF$).



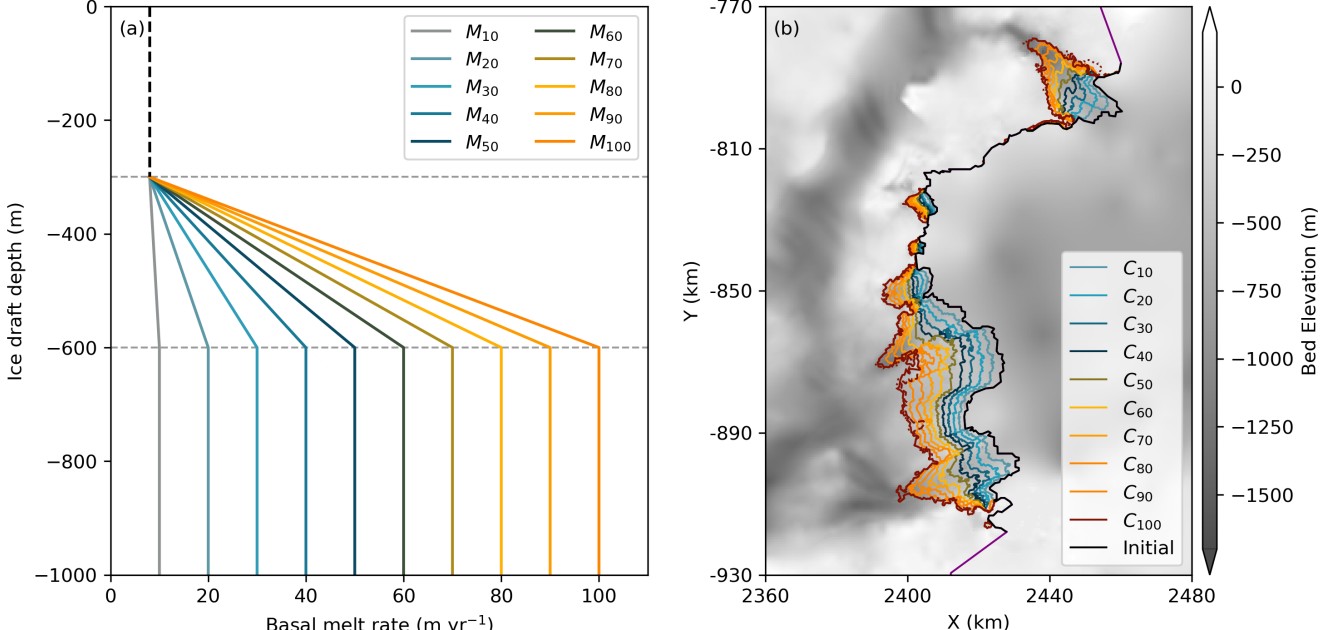

**Figure 4.** Overview of basal melt and calving perturbations. (a) Linear basal melt parameterisation. Vertical black dashed line represents constant $M_s$ across all experiments. Horizontal dashed lines denote shallow ($\geq$ -300 m) and deep ($<$ -600 m) ice drafts where changes in $M_d$ are applied. Note that the y-axis is truncated at -1000 m for visualisation; however, basal melt rates are applied across all ice drafts. (b) Final ice-front positions for all calving ($C_l$ and $C_m$) perturbations. Note that final ice-front positions are held constant following the perturbation period. The 100 % IFR perturbation varies depending on the friction law due to the specific grounding line location: Budd (solid), Schoof (dashed), and Weertman (dotted), although differences are very minor and 100 % IFR lines plot on top of each other in (b). The purple outline denotes the model domain boundary. Ice front positions are plotted over bed elevation from BedMachine v3 (Morlighem et al., 2020).

## 3    Results

We assess the results for each set of perturbation experiments individually. For each, we first consider the system response during the perturbation period and then the evolution following the perturbation period. We then consider the influence of the basal friction law during the perturbation period, followed by its influence on the evolution of the system after the perturbation period. The results of the observed mass loss perturbation experiments ($M_{Paolo}$, $M_{Davison}$, and $C_{obs}$) are presented in Sect. 3.1. Basal melt perturbation experiment results are shown in Sect. 3.2,

calving perturbation experiments in Sect. 3.3, and hybrid experiments in Sect. 3.4.





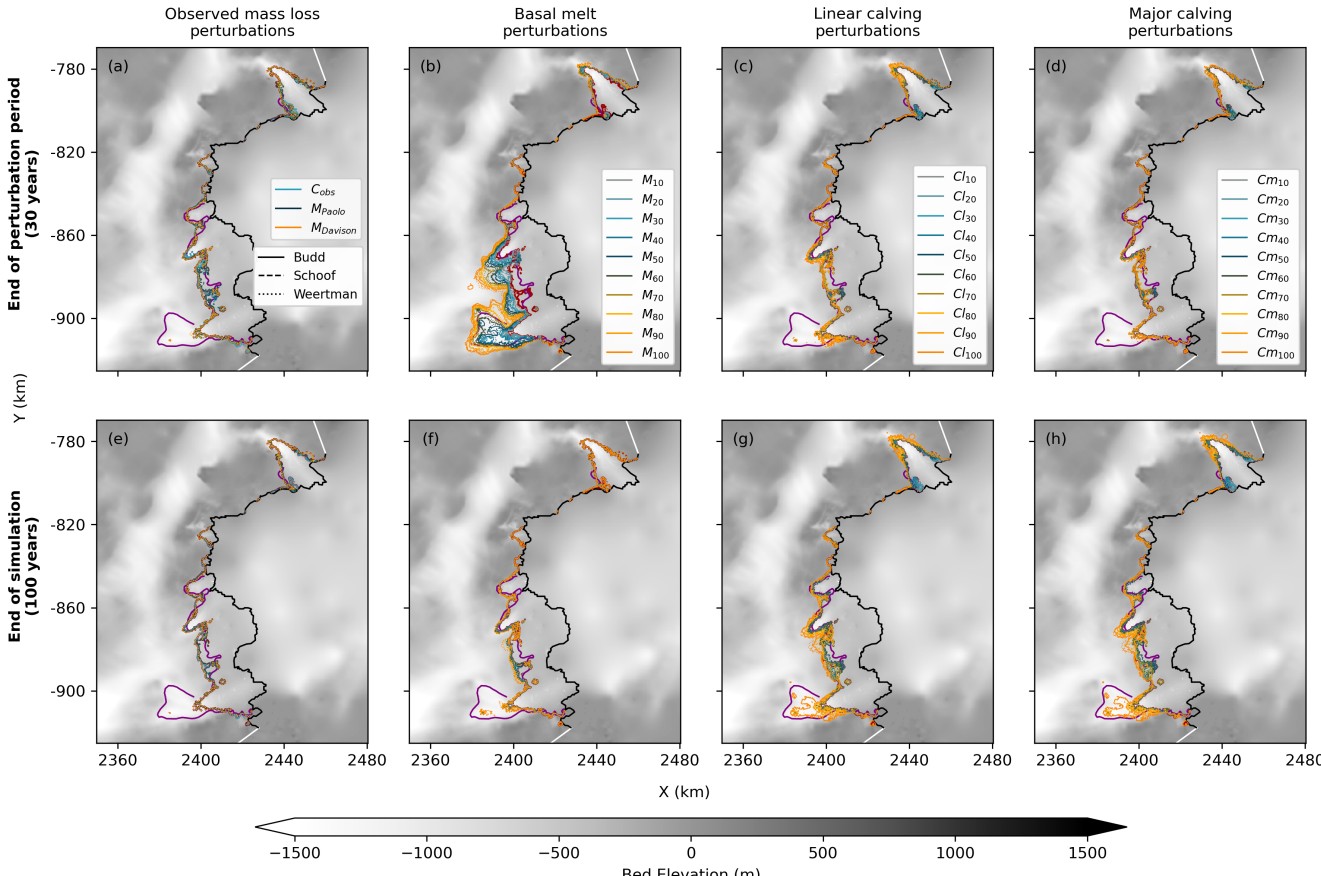

**Figure 5.** Grounding line positions across all perturbation experiments. (a)-(d) Grounding line positions at the end of the perturbation period (30 years). (e)-(h) Grounding line positions at the end of the simulation (100 years). (a) and (e) show observed mass loss simulations; (b) and (f) show basal melt perturbations; (c) and (g) show linear calving ($Cl$) perturbations; and (d) and (h) show major calving ($Cm$) perturbations. Grounding lines in red show the initial grounding line for each simulation. The 2020 observed grounding line (Picton et al., 2023) is shown in purple. Grounding lines are overlain on bed elevation from BedMachine v3 (Morlighem et al., 2020). The black line is the initial ice-front and white line is the domain boundary.



**Table 1.** Overview of perturbation experiment forcings. Values in parentheses denote the increments of applied forcings across experiments. $M_d$ = deep ice draft basal melt rate; $M_s$ = shallow ice draft basal melt rate. Subscripts used in experiment names for basal melt perturbations denote the corresponding $M_d$ value. Subscripts used in experiment names for the calving perturbations denote the corresponding IFR percentage. No specific experiment names are assigned to hybrid experiments.

| Perturbation Type | Experiment name | $M_d$ (m yr$^{-1}$) | $M_s$ (m yr$^{-1}$) | Ice-front retreat (%) |
|---|---|---|---|---|
| None (Control) | $Ctrl$ | 3 | 3 | 0 |
| | $C_{obs}$ | 3 | 3 | Observed |
| Observed Mass Loss | $M_{Paolo}$ | Observed | Observed | 0 |
| | $M_{Davison}$ | Observed | Observed | 0 |
| Basal Melt | $M_{10}$ - $M_{100}$ | 10 - 100 (10) | 8 | 0 |
| Calving (Linear) | $Cl_{10}$ - $Cl_{100}$ | 3 | 3 | 10 -100 (10) |
| Calving (Major) | $Cm_{10}$ - $Cm_{100}$ | 3 | 3 | 10 -100 (10) |
| Hybrid | – | 10 - 50 (10) | 8 | 20 - 100 (20) |

### 3.1 Observed mass loss perturbations

None of the observed mass loss experiments result in marked changes in $GL_{flux}$, $VAF$, or grounding line position at any time throughout the simulations (Fig. 5a and Fig. 5e; Fig. 6b). Most mass loss occurs directly from the ice shelf, rather than thinning or increased discharge of upstream grounded ice (Fig. 6c-d and Fig. S1). The $C_{obs}$ experiment resulted in greater mass loss than both the $M_{Paolo}$ and $M_{Davison}$ experiments (Fig. 6d).

During the perturbation period, the $M_{Paolo}$ and $M_{Davison}$ experiments show similar responses across all variables, except for $VAF$ where the $M_{Paolo}$ experiment shows an increase and the $M_{Davison}$ experiment shows a decrease (Fig. 6c). Negative basal melt rates (i.e. ice accretion) in the $M_{Paolo}$ experiment cause slight ice-shelf thickening close to the Vanderford Glacier grounding line, compared to widespread ice-shelf thinning simulated in the $M_{Davsion}$ experiment (Fig. S1). Each experiment ($C_{obs}$, $M_{Paolo}$, and $M_{Davison}$) shows a pattern of change in $VAF$ and ice volume that is generally consistent across friction laws, with differences in the magnitude of the change. For example, the $M_{Paolo}$ experiment shows the largest range of $VAF$ changes, from 2.8 - 7.6 Gt yr$^{-1}$ between the friction laws. Changes in $GL_{flux}$ are negligible for all experiments ($C_{obs}$, $M_{Paolo}$, and $M_{Davison}$), but there are small variations in $GL_{flux}$ (i.e. on the order of 0.2 Gt yr$^{-1}$) between the friction laws.

Following the perturbation period, all fields in the $M_{Paolo}$ experiment trend towards their initial state, irrespective of the friction law. By the end of the simulation, $VAF$ and ice volume remain elevated from their initial state due to thickening of grounded ice (Fig. S2). The $M_{Davison}$ experiment with the Budd friction law shows continued mass loss, while the Schoof and Weertman friction laws show a tendency for ice volume to trend towards its initial state (Fig. 6c-d). The different response of the Budd friction law between $M_{Paolo}$ and $M_{Davison}$ is attributed to more





widespread thinning of ice upstream of the grounding line (Fig. S2). The $C_{obs}$ experiment shows no increase in ice volume due to the persistence of the final retreated ice-front compared to the initial location (Fig. 6d).

Overall, the system shows a negligible response when perturbed with observed basal melt and calving events. There is minor variability between friction laws, and most notable changes in $VAF$ and ice volume result from disparities in basal melt estimates, particularly close to the grounding line.

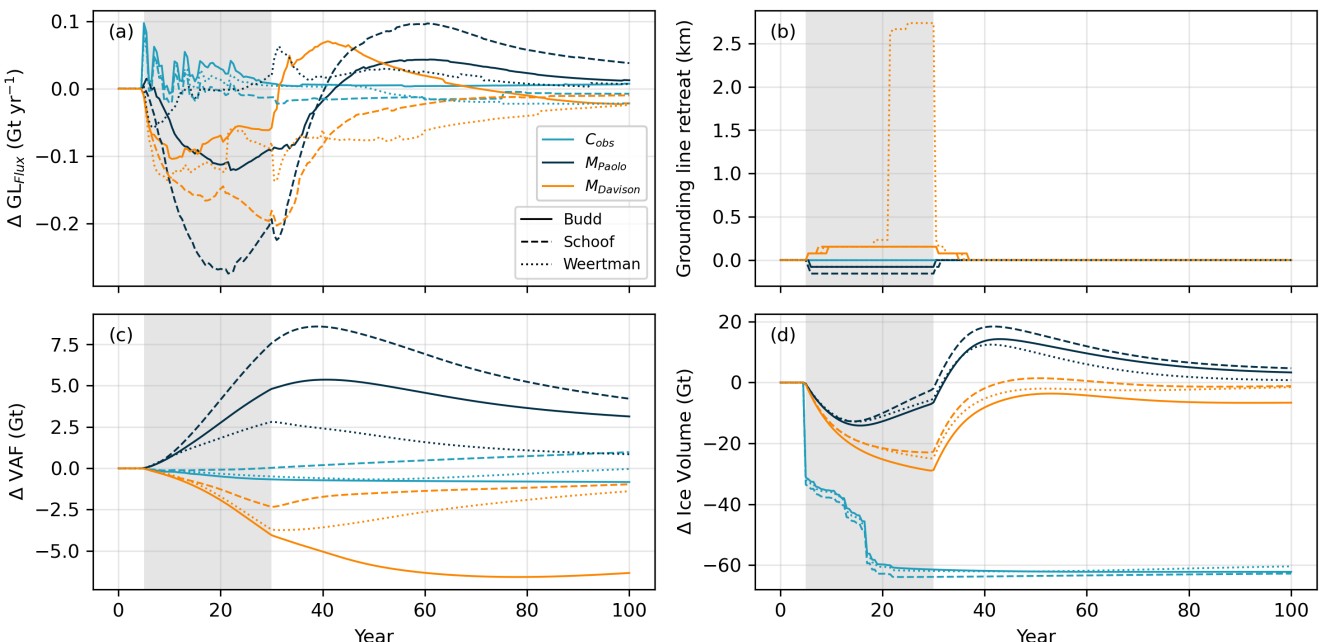

**Figure 6.** Results of observed mass loss perturbations. Relative change (compared to the $Ctrl$ experiment for each friction law) of (a) Grounding line flux, (b) Grounding line retreat along the central flowline shown in Fig. 2c, (c) Volume above floatation, and (d) Ice volume. Grey shaded area denotes the perturbation period.

## 3.2    Basal melt perturbations

Over the perturbation period, all basal melt experiments yield an increase in $GL_{flux}$, ice volume loss, and grounding line retreat (Fig. 5c and Fig. 7). Grounding line retreat into regions of deeper bed topography, where increased ice velocities dominate over general thinning of upstream ice, leads to increased $GL_{flux}$ (Fig. 7a). Ice volume loss and $GL_{flux}$ increase linearly with $M_d$ values $> 50$ m yr$^{-1}$. All experiments show a decrease in $VAF$ (Fig. 7g), consistent 245    with dynamic thinning of grounded ice upstream of the grounding line (Fig. S3a and Fig S4). For $M_d$ values $\leq 50$ m yr$^{-1}$, our experiments suggest maximum grounding line retreat of 9.7 km, while $M_d$ values $> 50$ m yr$^{-1}$ lead to grounding line retreat within the range of observed retreat (15.2 - 21.7 km).





Following the perturbation period, all experiments show grounding line re-advance, with the final grounding line position close to the initial location (Fig. 5f; Fig. 7d). The $VAF$ continues to reduce for a period of ~30 years,
consistent with continued thinning upstream of the grounding line (Fig. S3a). Total ice volume increases from the end of the perturbation period resulting from ice shelf thickening (Fig. S3d). The $GL_{flux}$ reduces as a result of persistent thinning upstream of the grounding line which is not offset by sufficiently large and sustained magnitude changes in velocity (Fig. S3a and Fig. S3g).

During the perturbation period, the pattern of system response is similar for all friction laws, but the magnitude
differs. The Weertman friction law generally yields the smallest change in $GL_{flux}$. For $M_d$ values $\leq 50$ m yr$^{-1}$, the Weertman friction law shows more extensive grounding line retreat than the other friction laws; however, for $M_d$ values $> 50$ m yr$^{-1}$ the Budd friction law causes more extensive grounding line retreat. Ice volume at the end of the perturbation period does not show a clear relationship with the choice of friction law, with each yielding a similar response.

Following the perturbation period, the choice of friction law plays a key role in the rate and magnitude of ice volume evolution (Fig. 7g and Fig. 7j). The Budd friction law consistently yields the smallest rate of ice volume increase due to higher magnitude dynamic thinning upstream of the grounding line following the perturbation period (Fig. S3a). The Weertman friction law consistently yields the highest rate of ice volume increase.

Overall, the system shows a similar pattern of response to all basal melt perturbations and all friction laws, with an
increased magnitude of response as $M_d$ increases. Values of $M_d > 50$ m yr$^{-1}$ are required to generate grounding line retreat close to the observed grounding line location which is much larger than current estimates of basal melt (Fig. 2).

### 3.3 Calving perturbations

Over the perturbation period, all calving experiments yield an increase in $GL_{flux}$ and ice volume loss, irrespective
of the calving mechanism ($Cl$ or $Cm$). Changes in $VAF$ are minimal, highlighting that mass loss predominantly occurs from floating ice in both $Cl$ and $Cm$ experiments (Fig. 7h and Fig. 7j). Major calving ($Cm$) experiments show an instantaneous response in $GL_{flux}$ (Fig. 7c) and ice volume (Fig. 7l) to calving events which occur at 12.5-year intervals (5, 17.5, and 30 years), while linear calving ($Cl$) experiments show a more gradual response (Fig. 7b and Fig. 7k). The magnitude of $GL_{flux}$ increase is negligible and no grounding line retreat occurs for all experiments with
$< 80$ % IFR (i.e. $Cl_{10}$-$Cl_{70}$ and $Cm_{10}$-$Cm_{70}$; Fig. 5c-d and Fig. 7e-f). Notable grounding line retreat only occurs towards the end of the perturbation period for the experiments with 100 % IFR (i.e. ice-shelf removal). The $Cl$ experiments show slightly earlier onset and marginally larger magnitude grounding line retreat compared to the $Cm$ experiments (Fig. 7e-f).







**Figure 7.** Results of basal melt and calving perturbations. Relative change (compared to the $Ctrl$ experiment for each friction law) of: (a-c) Grounding line flux, (d-f) Grounding line retreat along the central flowline shown in Fig. 2c, (g-i) Volume above floatation, and (j-l) Ice volume. Basal melt ($M$) perturbations are shown in the first column, linear calving ($Cl$) perturbations in the second, and major calving ($Cm$) perturbations in the third. Grey shaded area denotes the perturbation period. Purple horizontal line in (d-f) shows grounding line retreat along the central flowline shown in Fig. 2c, from the initial mean grounding line across all friction laws to the 2020 observed grounding line location (Picton et al., 2023)





Following the perturbation period, $GL_{flux}$ reduces, but remains elevated from its initial state for all experiments
with $\geq 80$ % IFR, due to increased velocities and dynamic thinning upstream of the grounding line (Fig. S3b-c and
Fig. S3h-i). The $Cl$ and $Cm$ experiments with $< 80$ % IFR show negligible additional change in ice volume or $VAF$,
while experiments with $\geq 80$ % IFR show continued ice volume loss, consistent with the persistent increased $GL_{flux}$.
Grounding line retreat continues for all experiments with $\geq 80$% IFR, with only minor retreat ($< 1$ km) for $Cl_{80}$ and
$Cm_{80}$, and a maximum retreat of ~9 km for the $Cl_{100}$ and $Cm_{100}$ experiments (Fig. 5g-h and Fig. 7e-f).

During the perturbation period, the pattern of the system response is similar for all friction laws, but the magnitude
differs. The Schoof friction law consistently yields the largest mass loss (Fig. 7k and Fig. 7j). The magnitude of
grounding line retreat is similar between all experiments, with the Weertman friction law leading to greater retreat
for experiments with 90 % and 100 % IFR. For $Cl$ and $Cm$ experiments with $\geq 80$ % IFR, the Schoof friction law
yields a greater $GL_{flux}$.

Following the perturbation period, $GL_{flux}$ reduces, but remains elevated for all Budd friction law experiments and
Schoof and Weertman friction law experiments with $\geq 80$ % IFR (Fig. 7b-c). The Budd friction law consistently
yields the highest $GL_{flux}$ at the end of the simulation. The Weertman friction law generally yields the largest ice
volume at the end of the simulation period, while the Budd and Schoof friction laws generate more varied results
(Fig. 7k-j). Continued grounding line retreat is generally consistent between all experiments, with the Schoof friction
law generating marginally larger retreat for experiments with $\geq 80$ % IFR (Fig. 5g-h and Fig. 7e-f).

The mechanism of calving (i.e. $Cl$ or $Cm$) primarily affects the response of the system over the perturbation
period, particularly changes in $GL_{flux}$. We compare the absolute cumulative change in $GL_{flux}$ between $Cl$ and $Cm$
experiments following large-scale calving events and at the end of the simulation period (i.e. at 5, 17.5, 30, and 100
years) to directly compare the influence of the calving mechanism on dynamic mass loss (Fig. 8). Figure 8 shows the
similarity of the absolute cumulative change in $GL_{flux}$ between $Cl$ and $Cm$ calving perturbations. After the initial
major calving event (5 years), the mean (i.e. across all experiments and friction laws) absolute cumulative change in
$GL_{flux}$ from $Cm$ experiments ($0.14 \pm 0.09$ Gt yr$^{-1}$) is several magnitudes larger than $Cl$ experiments ($3 \times 10^{-5} \pm$
$1 \times 10^{-5}$), primarily due to negligible changes in $GL_{flux}$ for $Cl$ experiments and larger magnitude instantaneous
changes for $Cm$ experiments. At the next major caving event (17.5 years), the mean absolute cumulative change in
$GL_{flux}$ from $Cm$ experiments ($2.35 \pm 1.65$ Gt yr$^{-1}$) is 14 % larger than from $Cl$ experiments ($2.06 \pm 1.38$ Gt yr$^{-1}$).
Subsequently, at the next major calving event (30 years) $Cl$ experiments yield a mean absolute cumulative change in
$GL_{flux}$ ($14.66 \pm 18.97$ Gt yr$^{-1}$) that is 92 % larger compared to $Cm$ experiments ($7.64 \pm 7.31$ Gt yr$^{-1}$). At the end
of the simulation period, the mean absolute cumulative change in $GL_{flux}$ from $Cl$ experiments ($119 \pm 196$ Gt yr$^{-1}$)
is $< 1$ % larger than that from $Cm$ experiments ($118 \pm 195$ Gt yr$^{-1}$).



Overall, while the calving mechanism ($Cl$ or $Cm$) affects the instantaneous response of the system, there is a similar response across $Cl$ and $Cm$ experiments over the 100 year simulation period. IFR of $\geq 80$ % is required to generate any notable system response which is much larger than observed IFR (Fig. 2).

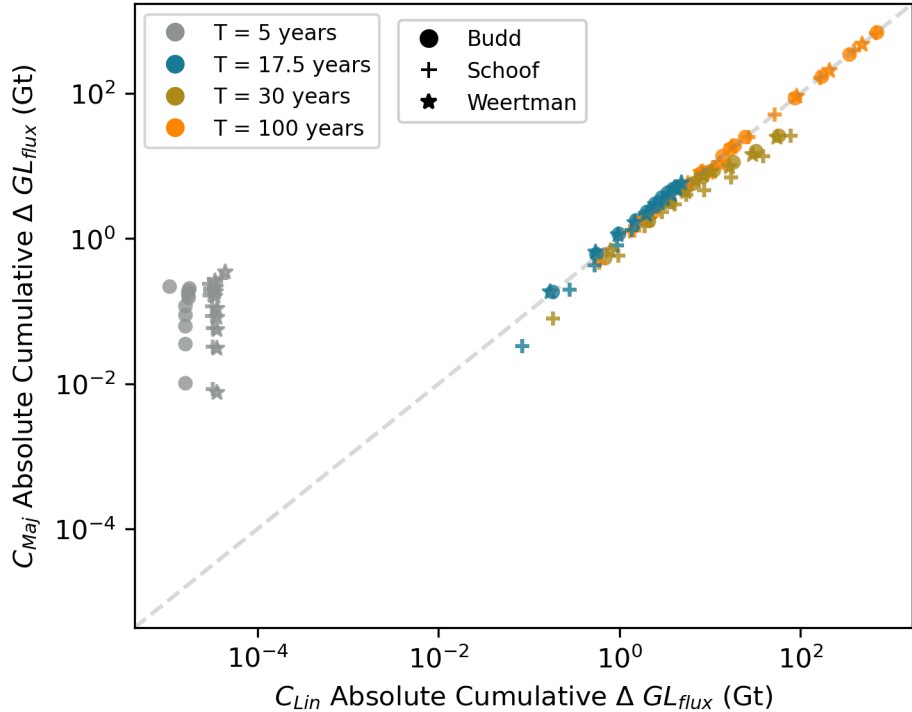

**Figure 8.** Comparison of absolute cumulative relative change (compared to the $Ctrl$ experiment for each friction law) in grounding line flux between $Cl$ and $Cm$ experiments at select timesteps throughout the simulation period.

### 3.4   Hybrid experiments

Section 3.2 and Section 3.3 show that $M_d$ values $> 50$ m yr$^{-1}$ are required to generate grounding line retreat close to

the observed position and calving processes yield negligible grounding line retreat until $\geq 80$ % IFR occurs. Here, we explore whether combined lower-magnitude basal melt and calving perturbations can generate a comparable system response. The addition of IFR to low magnitude basal melt processes has a negligible effect on $GL_{flux}$ or grounding line position until $\geq 80$ % IFR has occurred (Fig. 9), consistent with the effects shown in Sect. 3.3.

The relative change in $GL_{flux}$ due to IFR generally shows an inverse relationship with $M_d$ for experiments with $\geq$

80 % IFR (Fig. 9a-c). We attribute this to the relative increase in direct mass loss from the ice shelf decreasing with $M_d$, resulting in reduced buttressing forces. That is, ice shelf thickness is generally inversely related to $M_d$ such that less mass is removed over an equivalent area of calving as ice thickness decreases. For experiments with $\geq 80\%$ IFR,



the Schoof friction law generally results in the largest increase in $GL_{flux}$, while the Budd friction law generally yields the smallest (Fig. 9a-c). For example, for the hybrid experiment with $M_d = 50$ m yr$^{-1}$ and 100% IFR, the Schoof friction law results in a ~6 % greater influence (17 % increase in $GL_{flux}$) compared to the Budd friction law (~11 % increase in $GL_{flux}$).

The effect of including IFR on grounding line retreat shows no clear pattern, with the greatest retreat occurring for different $M_d$ values across different friction laws. The Schoof friction law generally results in more consistent and greater magnitude grounding line retreat across all experiments (Fig. 9d-f). However, none of the experiments yield grounding line positions close to the observed position by the end of the perturbation period (see absolute values shown in Fig. 9d-f).

While the addition of IFR to basal melt perturbations generally results in increased $GL_{flux}$ and additional grounding line retreat, the influence is insufficient to generate sufficient additional grounding line retreat required to match the observed grounding line position. Thus, as in the standalone basal melt experiments, high basal melt rates at depth (i.e. $M_d > 50$ m yr$^{-1}$) are required to simulate grounding line retreat similar to observed.

## 4   Discussion

This study examines the sensitivity of Vanderford Glacier to sub-ice shelf basal melt and ice-front retreat (IFR). We show that basal melt rates $> 50$ m yr$^{-1}$ at depth are required to generate grounding line retreat close to the present-day location over the perturbation period; approximately twice those of Paolo et al. (2022) and Davison et al. (2023). Importantly, our simulations show that the observed grounding line retreat at Vanderford Glacier is unlikely to result from calving, since grounding line retreat only occurs with $\geq 80$ % IFR; much larger magnitude IFR than has been observed. In the absence of basal melting at depth $> 50$ m yr$^{-1}$, even when the entire ice-shelf is removed ($Cl_{100}$ and $Cm_{100}$), the magnitude and rate of simulated grounding line retreat is not comparable to observations. For example, it takes more than 95 years for $Cl_{100}$ experiments to generate a maximum grounding line retreat of ~9 km. It is possible that the observed grounding line retreat could be in response to a major calving event prior to the satellite observational era; however, we believe this is unlikely due to the scale of event that would be required to generate the observed retreat, the absence of other evidence conducive to previous ice-shelf collapse (e.g. rapid changes in ice thickness and ice velocity), and the size of observed calving events in recent decades being of much smaller magnitude. Our results indicate that basal melting is the likely driver of recent grounding line retreat at Vanderford Glacier.

When forced with current observational estimates of calving and basal melt, our simulations do not generate grounding line retreat close to the present-day location over the perturbation period. This suggests that estimates of sub-ice shelf basal melt rates from satellite altimetry (Paolo et al., 2022; Davison et al., 2023) may substantially underestimate basal melt, particularly close to the grounding line. A possible cause of this underestimation is the assumption of





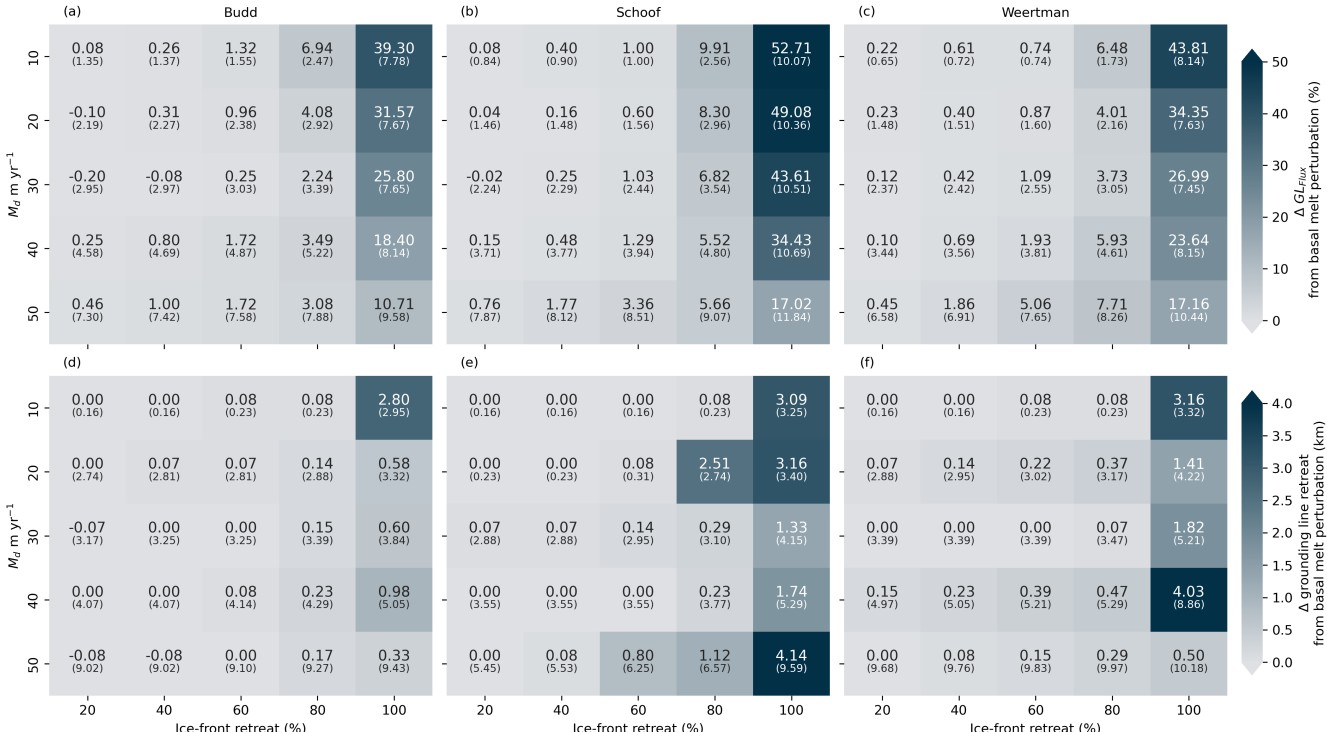

**Figure 9.** Influence of IFR on $GL_{flux}$ and grounding line retreat when combined with basal melt. (a-c) Change in $GL_{flux}$ and (d-f) Change in grounding line retreat. The colour scale and large font numbers display changes relative to the corresponding basal melt perturbation for each friction law, as per the colour scale label. Values in parentheses show the absolute relative change of each variable compared to the control experiment. For example, the Budd friction law experiment with $M_d = 40$ m yr$^{-1}$ and 100 % IFR shows that including IFR produces an 18.40 % increase in $GL_{flux}$, compared to the $M_{40}$ basal melt perturbation, and the combined experiment yields a 8.14 Gt yr$^{-1}$ increase in $GL_{flux}$ relative to the Budd control experiment.

hydrostatic equilibrium (i.e. that ice is freely floating across the entire ice shelf) in methods to derive basal melt from satellite-altimetry. In regions with steep ice thickness gradients and at shear margins, the validity of this assumption is challenged due to high strain rates and associated bridging stresses (e.g. Wearing et al., 2021; Chartrand and Howat, 2020, 2023; Dow et al., 2024). Chartrand and Howat (2023) investigate the effects of uncertainties in estimates of hydrostatic thickness on basal mass balance estimates for Antarctica, finding the Vincennes Bay / 

Underwood ice shelves have some of the largest uncertainties in hydrostatic thickness-derived basal melt rates in East Antarctica. These hydrostatic thickness uncertainties result in large uncertainties in basal mass balance estimates across Vincennes Bay, influenced largely by spatial changes in the strain rate across the ice shelves. This highlights the need for improved estimates of basal melt across Vincennes Bay ice shelves, and the need for in situ measurements (e.g. geophysical or oceanographic) to constrain remotely-sensed estimates.





The melt rate parameterisation used in this study is a highly simplified representation of basal melting. While it captures the depth dependence of melt rates consistent with mCDW-driven melt, it neglects the role of subglacial discharge and ocean dynamics which are known to impact melt rates in the neighbouring Totten Glacier ice shelf (Gwyther et al., 2023; Xia et al., 2023) and may also play an important role in Vincennes Bay ice shelves (Jacobs et al., 1992; Silvano et al., 2016). Importantly, spatial and temporal variations in thermocline depth, which are known

to have implications on mode 2 melting (e.g. De Rydt et al., 2014; Dutrieux et al., 2014; Hoffman et al., 2019), are not captured by this melt rate parameterisation. Here, we select a thermocline depth based on the depth at which mCDW is predominantly found in Vincennes Bay (Ribeiro et al., 2021). Previous studies have shown a dynamic subglacial hydrology network beneath the Totten Glacier drainage basin (Dow et al., 2020), and ocean modelling of Totten ice shelf cavity shows that subglacial freshwater discharge from this network can enhance basal melting by

over 3% across the ice shelf, with localised increases of 25 % - 30 % in regions close to the grounding line (Gwyther et al., 2023). Although Dow et al. (2020) does not include the whole Vincennes Bay hydrological catchment in their analysis, there is evidence for an extensive subglacial hydrology system across the Aurora Subglacial Basin (Wright et al., 2012), potentially with strong connectivity between the Totten and Vanderford upstream catchments. Hence, similar localised convective plumes resulting from buoyant discharge of freshwater that enhance basal melting in the

Totten Glacier ice shelf cavity may also play a strong role in basal melting in the Vanderford Glacier ice-shelf cavity.

More complex basal melt parameterisations (e.g. PICO/PICOP and the ISMIP-6 protocol; Reese et al., 2018; Pelle et al., 2019; Jourdain et al., 2020) rely on temperature and salinity forcing from regional ocean model output. However, known disparities in offshore bathymetry in Vincennes Bay may limit the accuracy of regional ocean models here, since they may not correctly represent local ocean conditions that may arise from bathymetric features. Specifically,

the presence of the deep marine trough mapped offshore the Vanderford Glacier (Commonwealth of Australia, 2022) is absent in continental bathymetry estimates (e.g. BedMachine V3 ; Morlighem et al., 2020) commonly used within ocean models. This bathymetric feature could have important implications on ocean dynamics, potentially providing a pathway for mCDW into the Vanderford Glacier ice shelf cavity. Thus, without this feature, ocean models may underestimate basal melt in the Vanderford Glacier ice shelf cavity. Similarly, uncertainties in bed topography

upstream of the grounding line could have implications on our simulated grounding line retreat, with deeper bed topography potentially resulting in more rapid or extensive grounding line retreat occurring with lower basal melt rates. This highlights the need for improved bathymetric mapping across Vincennes Bay to support more reliable ocean modelling and basal melt rate estimates, as well as additional geophysical data acquisition over grounded ice to improve bed topography estimates.

One of the consequences of the simplification in our melt parameterisation is a potentially spurious reconfiguration of ice flow. That is, local changes in ice velocity across Vanderford Glacier ice shelf show a decrease in the strength of the western shear margin (Fig. S5) indicative of ice-flow reconfiguration based on changes in the driving stress. While shear margin migration has been found to induce grounding line retreat and promote mass loss (Feldmann et al.,



2022; Lhermitte et al., 2020), observational records of ice velocity at Vanderford Glacier do not show this migration.
We attribute this to our simulated melt rates in the region to the west of Vanderford Glacier likely being too high. We note that the ice-flow trends towards its original configuration once the basal melt perturbation is removed and the ice shelf re-thickens.

Joughin et al. (2021) show that simulated mass loss at Pine Island Glacier exhibits only minor sensitivity (<6 %) to melt distribution and depends linearly on the total melt. We test whether this is also the case for Vincennes Bay by
running four additional experiments with constant melt rates of 10, 20, 30, and 40 m yr$^{-1}$ across all floating ice. We find that notable grounding line retreat towards the observed location requires constant melt rates of $\geq 30$ m yr$^{-1}$ (Fig. S6). However, for melt rates $> 10$ m yr$^{-1}$, large portions of the ice-shelf are completely removed (Fig. S6). This behaviour is not supported by observations, which may suggest that Vanderford Glacier grounding line retreat is driven by elevated basal melt at depth, consistent with observed mCDW intrusions within Vincennes Bay primarily
located at 300 - 600 m deep (Ribeiro et al., 2021; Herraiz-Borreguero and Naveira Garabato, 2022).

Ocean modelling of future ice shelf basal melt around Antarctica shows that basal mass loss could increase between 33-83% in the Australian sector of East Antarctica by the end of the century (Naughten et al., 2018), depending on the future climate forcing scenario. Increased future mass loss projections suggest that continued grounding line retreat, from reduced buttressing forces, may be observed at Vanderford Glacier over coming decades, consistent
with findings from Sun et al. (2016). The current dominance of mCDW-driven melt may be reduced as Antarctica Surface Water becomes more prevalent with a reduction in future sea ice cover (Naughten et al., 2018). In turn, this could lead to mode 3- (i.e. surface-) dominated melting in the region in the future. Our experiments with a constant melt rate across the entire ice shelf suggest that a regime shift towards mode 3-dominated melt could increase mass loss from Vanderford Glacier as large portions of the ice shelf are removed (Fig. S6), and may lead to accelerated
grounding line retreat as buttressing forces are reduced.

Given the negligible effects of observed calving on simulated ice dynamics and grounding line position (i.e. experiment $C_{obs}$), it is likely that observed calving events have only removed passive ice (Fürst et al., 2016) from the Vincennes Bay calving front. To confirm this, we compute the maximum buttressing field following Fürst et al. (2016) using the initial velocities at the beginning of our experiments (Fig. 10). The observed calving extent (i.e. Greene et al., 2022)
does not extend beyond the 0.4 passive shelf-ice threshold (Fürst et al., 2016) for the Vanderford Glacier ice shelf. Our calving perturbation experiments suggest IFR $\geq 80$ % is required to initiate non-negligible changes in ice dynamics at Vanderford Glacier. This behaviour is generally consistent with findings from Mitcham et al. (2022) where an equivalent of ~90 % IFR of the Larsen C ice-shelf is required to generate a ~50 % increase in the grounding line flux in their idealised calving experiments. Along the main trunk of Vanderford Glacier where most ice discharge occurs,
the extent of 80% IFR is similar to the 0.8 instantaneous discharge threshold contour, as defined for Vanderford Glacier by Fürst et al. (2016) (Fig. 10).



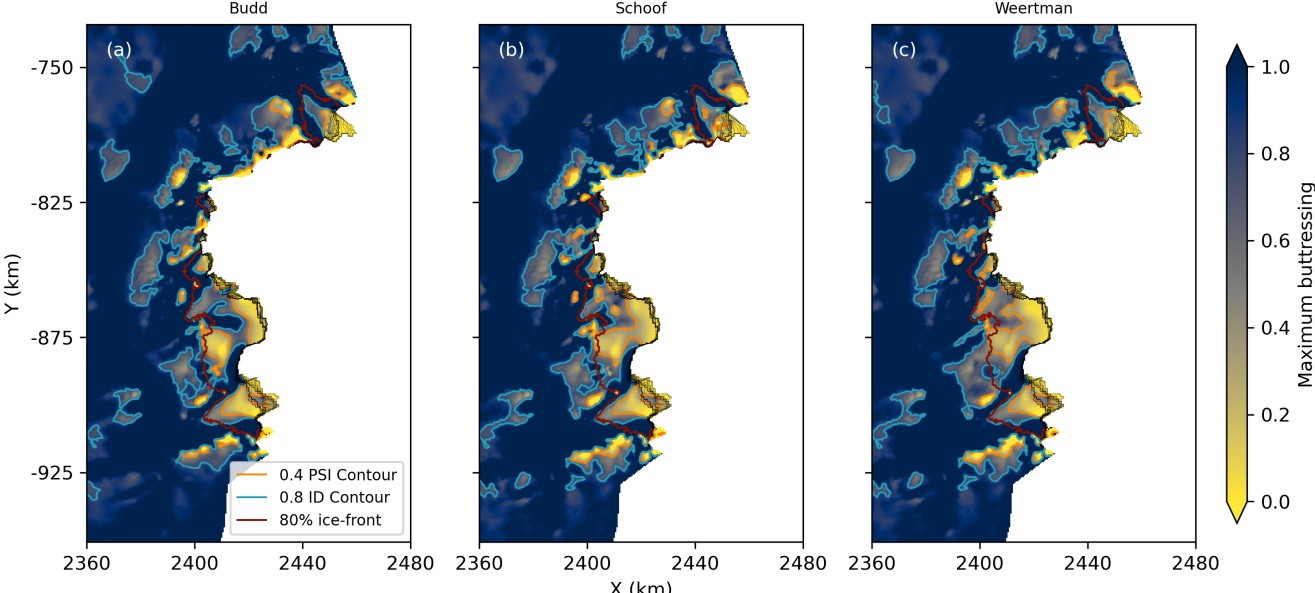

**Figure 10.** Maximum buttressing field calculated from initial velocities at the beginning of each perturbation experiment for each friction law (a) Budd, (b) Schoof, and (c) Weertman. Black lines display the extent of the observed ice-fronts from Greene et al. (2022). Maximum buttressing values are truncated between 0 and 1.

Our simulations show that the Vanderford Glacier system responds rapidly to calving events, and the response is generally independent of the process by which calving occurs (i.e. edge-wasting or major calving events). This may in part be due to the simplified implementation of calving used in our simulations. That is, we have used a level-set method to prescribe the location of the ice-front at each time-step (Sect. 2.3), which does not capture the influence of transient changes in stress and strain fields across the ice-shelf that may be better represented in more physically based calving laws (e.g. eigencalving or von Mises calving laws ; Wilner et al., 2023). Furthermore, we hold the ice-front at the final location (Fig. 4b) following the perturbation period while in reality, additional re-advance of and calving from the ice-front would occur. As such, our calving perturbations represent the worst-case impact of calving with a persistent retreated ice-front, such that the actual impacts of calving on grounding line retreat and mass loss are likely even smaller.

Our results show a varied response to different friction laws across the perturbation period and the whole simulation run, as well as between different variables we assess. In general, the Weertman friction law yields the smallest change in $VAF$ across the simulation period, and the Budd friction law yields the largest changes. The magnitude and upstream extent of grounded ice thickness changes are largest for the Budd friction law (Fig. S4). The behaviour of the Budd friction law is attributed to the relationship between basal shear stress, effective pressure, and sliding velocities (Joughin et al., 2019; Brondex et al., 2017, 2019). That is, in the Budd friction law, the basal shear stress is



linearly proportional to the effective pressure such that as grounded ice thins, the effective pressure is lowered, basal shear stress is reduced, and sliding increases (Carr et al., 2024). This is not the case in either the Weertman friction law (where there is no explicit representation of basal water pressure) or the Schoof friction law (where effective pressure is controlled by cavitation) (Joughin et al., 2019).

Our finding of the Budd friction law yielding the largest changes in $VAF$ is in agreement with previous studies for both idealised experiments (Brondex et al., 2017) and on simulations of the Amundsen basin (Brondex et al., 2019). Despite variability in the magnitude of the response between different friction laws, our simulations show a similar response and trajectory throughout the simulation. This is similar to findings from Barnes and Gudmundsson (2022) who show general consistency across their simulations of the Amundsen Sea Embayment for up to a century, suggesting that decadal- to centennial-scale simulations may be insensitive to the choice of friction law. We recognise that the choice of sliding law parameters are also shown to have important implications at similar timescales (Brondex et al., 2017, 2019; Barnes and Gudmundsson, 2022); however, assessing the sensitivity of the Vincennes Bay region to a full-suite of basal friction representations is out of scope for this study. Importantly, the presence of an active subglacial hydrology across the Aurora Subglacial Basin (Wright et al., 2012) likely means that these simple parameterisations of basal sliding will not represent the complex interactions at the ice-bed interface and that coupled subglacial hydrology and ice sheet model simulations are required to better understand the influence of basal sliding on the evolution of the Vincennes Bay region.

## 5 Conclusions

In this study, we explore the relative importance of basal melt and calving on grounding line retreat at Vanderford Glacier, and assess whether current estimates of mass loss can explain observed grounding line retreat. We found that basal melt rates $> 50$ m yr$^{-1}$ at depth are required to generate grounding line retreat close to observations over the 25-year perturbation period. Although this may be an overestimation of actual basal melt rates required to drive observed grounding line retreat (due to the simplified basal melt parameterisation used in this study), our findings show that it is likely ocean-driven melt that is responsible for recent observed grounding line retreat, and not calving. Simulations using observational datasets of basal melt and calving do not result in grounding line retreat close to observations over the perturbation period, suggesting that satellite altimetry datasets may substantially underestimate melt at Vanderford Glacier. Our calving experiments show that complete ice-shelf removal is required to generate grounding line retreat similar to observed; however, ice-shelf collapse has not been observed at Vanderford Glacier, indicating that calving is likely not responsible for observed grounding line retreat. By considering multiple friction laws, we show that the system response is not very sensitive to the choice of friction law over the 25-year perturbation period, but the subsequent evolution of the system is more dependent on the choice of friction law, highlighting the need for coupled subglacial hydrology and ice sheet model simulations to better understand changes at the ice-bed interface.



The findings of this study motivate further investigation into the use of satellite-derived basal melt rates (particularly close to the grounding line where the assumption of hydrostatic equilibrium may be challenged) to quantify mass loss and understand the relative importance of different mass loss drivers around Antarctica. Our study highlights the need for in situ measurements of basal melt to help constrain remotely-sensed datasets, particularly for small dynamic systems such as Vanderford Glacier. Changes at Vanderford Glacier are known to have broader implications across the Aurora Subglacial Basin due to the close relationship between Totten and Vanderford Glaciers' flow configurations. Therefore, our improved understanding of processes driving recent change at Vanderford Glacier promotes further consideration of potential large-scale changes across the Aurora Subglacial Basin in the coming decades to centuries.

*Code and data availability.* All of the datasets used in this study are publicly available. We use version 4.22 (Revision 27903) of the open-source ISSM software which is freely available at https://issm.jpl.nasa.gov/download/. Datasets used to initialise the model are available via the corresponding articles cited in this paper. Namely, ice geometries and bed topography from BedMachine V3 (Morlighem et al., 2020, https://nsidc.org/data/nsidc-0756/versions/3), surface velocities from MEaSUREs v2 (Rignot et al., 2017, https://nsidc.org/data/nsidc-0484/versions/2), mean annual basal melt rates from Paolo et al. (2022) (http://its-live-data.s3.amazonaws.com/height_change/Antarctica/Floating/ANT_G1920V01_IceShelfMelt.nc) and Davison et al. (2023) (https://zenodo.org/records/8052519). Observed grounding line positions from Picton et al. (2023) are available from the ENVEO cryoportal (cryoportal.enveo.at), entitled "GLL, Vincennes Bay, Antarctica, 1996-2020". Observed ice-front positions from Greene et al. (2022) are available from https://github.com/chadagreene/ice-shelf-geometry. Processed model output timeseries and associated scripts to recreate figures included in this paper are available at https://doi.org/10.26180/26170102.

*Author contributions.* LB conceived and led the study, conducted the ice sheet modelling and analysis, and prepared the manuscript. FSM and JB provided guidance on the ice sheet modelling methodology and analysis. All authors contributed to discussions and provided comments on the manuscript.

*Competing interests.* FSM is a member of the editorial board of *The Cryosphere*. All other authors declare that they have no competing interests.

ther geographical representation in this paper.





*Acknowledgements.* We thank Jason Roberts for insightful discussions throughout this study, and Allison Chartrand for insightful discussion which helped to improve the manuscript.

*Financial support.* This research has been supported by the Australian Research Council (ARC) Special Research Initiative (SRI) Securing Antarctica's Environmental Future (SR200100005). FSM and RJ were supported by ARC Discovery Early Career Research awards DE210101433 and DE210101923, respectively. LB was supported by the Monash Graduate Scholarship (MGS) and Monash International Tuition Scholarship (MITS). This research was undertaken with the assistance of resources from the National Computational Infrastructure (NCI Australia), an NCRIS enabled capability supported by the Australian Government.




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
