# Peer review of "Assessing the sensitivity of the Vanderford Glacier, East Antarctica, to basal melt and calving"

_EGUsphere, 2024_

## Referee Comment (RC1)

**Comment on egusphere-2024-2060**

Benjamin Getraer

September 2024

**1 General Comments**

This manuscript contains valuable information and impressively comprehensive experimental results for guiding future modeling, data collection, and interpretation of Vanderford Glacier. However, currently the takeaways are not sufficiently clear, certain aspects of the methodology need to be more clearly explained and defended, and in general the manuscript needs to be edited for clarity and some technical errors.

The results here substantially contribute to the general understanding of Vanderford Glacier retreat, showing that observed calving has occurred in passive, non-buttressing ice, and did not contribute to the observed retreat rates. The authors' conclusion that basal melting likely is responsible for the observed grounding line retreat at Vanderford Glacier is important and well supported by their sensitivity testing, which suggests that high but plausible basal melt rates can reproduce observed retreat.

The authors interpret that because altimetry based estimates of melt rates are too low to reproduce observed grounding line retreat in their model, these observational datasets may be incorrect in this region. This conclusion may be valid but ought to be supported and discussed more quantitatively—two ways this analysis could be further improved are suggested in Specific Comments.

As the manuscript discusses in a few places, there are other factors which play a significant role in the simulated grounding line dynamics, and additional observational constraints which would improve future modeling efforts, beyond the melt rate itself. The presentation of these other factors can be improved, as currently the emphasis on the melt rate as a source of error and uncertainty in the abstract and conclusion does not reflect the more balanced discussion section. I offer more specific comments on balancing the discussion of sources of error in Specific Comments.

Some of the scientific methods and assumptions need to be more clearly outlined and/or defended. These are also addressed in Specific Comments.

**2   Specific Comments**

**2.1   Abstract**

As written, the abstract suggests that both observed melt rates and observed calving rates are insufficient to drive observed grounding line retreat (lines 8–11), and then goes on to state that the grounding line retreat was likely driven by melting and not calving (lines 11–13). This structuring is confusing, and I think do not clearly communicate the most important findings of the paper.

The manuscript suggests/concludes later in the text (lines 421–426) that it is physically impossible for the observed calving to exert any significant effect on the grounding line, which is supported by their perturbation experiments and by their calculation of the maximum buttressing field. I think this is a much more clear finding by which to reject the influence of ice calving on the observed grounding line retreat, as it stands for itself and does not require further interpretation. In contrast, the statement in the abstract that "...calving experiments suggest that $> 80\%$ ice-front retreat—well in excess of the observed ice-front retreat since 1996—needs to occur to generate grounding line retreat similar to observations," still left me unsure whether the observed ice front retreat could account for some of the grounding line or not. This statement also sets up a dichotomy with the previous parallel statement on melt rates, which creates the impression that they may be equally insufficient to drive the observed retreat. In this context, the concluding statement (lines 12–13), that "retreat...is likely to be dominated by basal melt, with an almost negligible contribution from calving," appears arbitrary and unsubstantiated.

A re-framing of the abstract in a way that more clearly expresses the findings which support its conclusions would function as more accurate and less confusing summary of the paper.

**2.2   Interpretation of "low" melt rate observations**

The authors suggest that observational datasets of melt rates calculated from satellite altimetry may be incorrect in this region. Two suggestions are made to address and support this claim:

1. The primary cause of uncertainty in melt rate estimations that is cited by the authors is that portions of the ice shelf may be floating non-hydrostatically. This source of bias and uncertainty has been independently estimated in the Vincennes Bay region by Chartrand and Howat (2023, whom the authors cite) at $-0.8 \pm 12.7$ m w.e. a$^{-1}$ (mean and standard of deviation, which the authors do not directly cite). Whether the further end of this range, which suggests that melt could be underestimated by up to 13.5 m w.e. a$^{-1}$ (or 14.7 m a$^{-1}$ in terms of ice thickness), is sufficient to bridge the gap their model reveals between observed melt rates and grounding line retreat could be addressed directly by the authors.

2. In Figure 4a–b, the authors present the "Mean annual basal melt rate derived from satellite altimetry" from (Paolo et al., 2022) and (Davison et al., 2023), respectively. Given that these figures show dramatically different spatial patterns of basal melt/accumulation rates, a quantified measure of uncertainty in altimetry-observed melt rates could be estimated by calculating the variance in melt rates between these datasets which have 20 years of overlapping data. Comparing the variance in melt rate estimates between these two parallel datasets with their results could better inform their discussion of observational uncertainty, and how observed melt rates compare to their inference of very high melt rates at depth.

**2.3 Other sources of uncertainty**

An important finding of this manuscript is that current estimates of melt rates from the observational record are insufficient to reproduce the observed extent of grounding line retreat in the numerical simulations presented here. Extensive testing of potential sources of error in the model which could yield these results is out of the scope of the experiments which this manuscript presents, but the treatment of sources of error ought to be presented evenly in the abstract and conclusion. The authors highlight that bed topography has uncertainties which could exceed 500 meters in the region upstream of the grounding line. The observed melt rates are introduced as perturbations to a steady state spin up which may not share a similar geometry to the real historical ice sheet. If there is reason to conclude that uncertainty in melt rates could be a more important source of error than uncertainty in bed topography, or initial ice sheet state, in the grounding line dynamics of this model of Vanderford, then there should be clear evidence presented supporting that argument.

Additionally, while the authors suggest in-situ measurement of melt rates is an important future step to take (lines 363 and 484), this suggestion is not clearly supported by their results or discussion. The expense, difficulty, and lack of spatial and temporal coverage associated with direct in-situ measurements of basal melt rates make it unclear how such measurements would actually be incorporated into their model or contextualized. Simply put, a few measurements in a vast system that are very high or very low or in between do not necessarily help to characterize the behavior of the system at a useful scale. In comparison, other suggestions put forward in the discussion section are much better supported by the results and discussion of the manuscript: improved estimates of ice shelf geometry to constrain the existing, ice-shelf wide, remote sensed data which are already usable in the existing model; improved bathymetry and ocean state measurements around the ice shelf to constrain ocean circulation models and estimates of heat flux into the ice shelf cavity; improved bed topography beneath grounded and floating ice to constrain the ice sheet geometry and grounding line dynamics.

**2.4  Scientific Methods and Assumptions**

1. Lowering and smoothing of the bed underneath the ice shelf (lines 86–91). A more clear explanation of what happens in the models if these adjustments are not made would better allow the reader to judge the validity and possible bias introduced by this adjustment. The statement that "this approach may limit any grounding line advance across model simulations" leaves unanswered the question of what effect such a bias could have. If it does prevent grounding line advance beyond the initially imposed geometry, it may obscure a tendency of the model to advance in the spin-up stage.

2. The use of a 500 year spin-up steady state from which the perturbations are conducted could be introduced and defended much more clearly. My understanding from reading the manuscript is that there is an implicit assumption that the forcings which caused grounding line retreat over the observational record ought to be able to cause similar grounding line retreat when introduced as a perturbation from the 500 year steady state model. If this interpretation is correct, it should be much more clearly stated in the manuscript. Importantly, the state of the ice domain (i.e. thickness, velocity, strain rates) after the 500 year spin-up in comparison to the initialization fields from observations should be shown somewhere in the main text or in the supplement. Any significant differences between the model state before and after the spin up should be discussed in the manuscript as to how those differences should be interpreted, and how they may impact the results.

**2.5  Other comments**

**line 8**   "…instead, basal melt rates in excess of 50 m yr$^{-1}$ at the grounding line…." It is unclear how much more 50m yr$^{-1}$ is than the observed rates, making this statement vague and hard to interpret.

**lines 49–50**   The phrasing of this idea implies that we should not expect high local variability in melt rates, even though that is quite common. I agree that high melt rates nearby, forced by similar ocean water, supports the plausibility of similarly high melt rates under Vanderford, but only in the context of other evidence. The disparity between observed Totten and Vanderford melt rates alone, without the context of these results, is not necessarily surprising at all. This idea might be better suited for the discussion section.

**lines 55–60**   These sentences are not clearly connected to the introduction and seem at least partially related to or repeated in the discussion.

**Figure 2**   The Totten insets are blocking the Underwood ice shelf which is otherwise visible in the other figures. Because the colorbar is scaled to the higher melt rates on Totten, it is hard to actually interpret the range of observed

melt rates on Vanderford. Contours, labels, or different color limits could be better. Additionally, this figure does not make clear what the spatial extent of the observed melt rates are: does the observed melt rate field evolve in time with the ice shelf cavity to follow the grounding line retreat? Or are they only provided for this spatial extent (of the unretreated grounding line)?

**line 69–70** "...can current estimates of basal melt and ice-front retreat...." This sentence confused me because it sounded like the experiments would only use observed melt from a single snapshot in time (current estimates). The observed melt rate datasets and ice front retreats span at least 25 years of the observational record.

**line 106–109** The quantities here should be given units similar to lines 111–112.

**line 166** "...mean annual basal melt rates...." This seems to refer to an average across each year resulting in a different "two-dimensional mean annual melt rate field" (line 145) for each year. If that is the case, that should be clarified. Similarly, clarification is needed for the "Mean annual basal melt rate" fields shown in Figure 2a–b. My assumption in Figure 2 was that this is a two-dimensional field which forms the average per-year melt rate across the entire observational record (i.e. not just a single year). More explanation is needed to interpret what the mean annual basal melt rate fields actually are, and whether they are averaged across the whole record or if there is a different field for each year.

**Figure 5** Elevation shown is from BedMachine—is this showing the elevation with the lowering and smoothing of the bed? It would be preferable to see the bed elevation as it is actually implemented in the model.

**lines 217–218** How different is the flux across the grounding line in the spun up steady state from the actual observations used to initialize the model?

**Figure 6b** Why does the $M_{Davison}$–Weertman grounding line retreat so far and so suddenly? This retreat does not seem to be reflected in $\Delta$VAF (Figure 6c) or in the map (Figure 5a).

**lines 339–340** The melt rates required to drive grounding line retreat in the model are twice those of the observational data: This could be a clarifying result to add to the abstract, as it contextualizes what 50 m yr$^{-1}$ means in relation to observations.

**line 344** This struck me as a key finding: if removing the entire ice shelf relatively suddenly is actually not nearly enough to drive the rate of retreat observed, then it seems that melt rates needed to remain high at depth close to the grounding line as the ice cavity evolved.

**lines 417–420** Mode-3 melting is associated with melting close to the ice fronts, generally in passive zones which contribute little to buttressing (Adusumilli et al., 2020). In this model, removing the front portions of the ice shelf entirely does almost nothing. Why would this type of melting be worse than melting at depth dominated by CDW?

**line 432** "Our simulations show that the Vanderford Glacier system responds rapidly to calving events." This was confusing to me, as Figure 7 shows almost no response to most calving, and if anything a delayed, ongoing response to large calving.

**lines 474–476** This could include the results that observed calving have had no effect. The phrasing here is slightly convoluted and confusing.

**lines 481–488** The final paragraph of the conclusion includes vague phrasing and does not highlight a strong implication of the results. Overall the conclusion repeats a lot of the discussion. My takeaway from the manuscript overall is that there is an unresolved incompatibility between modeled dynamics, observed forcings, and observed dynamics. A clear and succinct outline of those incompatibilities, and a stance taken on what the implication of those incompatibilities are until they are resolved would be enlightening and interesting.

**3 Technical Corrections**

**Figure 6c–d** y-axes are labeled as changes in volumes with units of Gt. The labels should be corrected.

**References**

Adusumilli, S., Fricker, H. A., Medley, B., Padman, L., and Siegfried, M. R.: Interannual variations in meltwater input to the Southern Ocean from Antarctic ice shelves, Nature geoscience, 13, 616–620, 2020.

Chartrand, A. M. and Howat, I. M.: A comparison of contemporaneous airborne altimetry and ice-thickness measurements of Antarctic ice shelves, Journal of Glaciology, p. 1–14, https://doi.org/10.1017/jog.2023.49, 2023.

---

## Referee Comment (RC3)

**Title**: Assessing the sensitivity of the Vanderford Glacier, East Antarctica, to basal melt and calving
**Authors**: Bird et al. (2024)
**Journal**: The-Cryosphere
**Reviewer**: T. Pelle (tpelle@ucsd.edu)

**Overview**:
In this manuscript, Bird et al. present an ice sheet modeling study that investigates the role of ice shelf basal melting and ice shelf calving in driving observed retreat of Vanderford Glacier in East Antarctica. The author's primary conclusions are that ice shelf basal melting (>50 m/yr) is the primary forcer of this observed retreat and that currently available satellite estimates of basal melting are insufficient to drive the magnitude of retreat observed over the past ~25 years. On the other hand, ice shelf calving had minimal impact on the dynamic retreat of Vanderford Glacier. Overall, I found this paper to be a pleasure to read! The ice sheet model and experimental set-up are meticulously described and the writing is free from any grammatical errors. Furthermore, I find that the conclusions are generally well supported by the results and that the authors make sure to highlight major sources of uncertainty. I also believe that the results of the paper will be important for future ice sheet modeling studies in this region, as this paper presents important constraints on the choice of ice sheet friction law as well as which forcing mechanisms to prioritize in the modeling of Vanderford Glacier. Below, I include suggestions that could improve the presentation and quality of the manuscript, but they are generally minor and should be relatively easy for the authors to address. Once this is done, I would be very supportive of this manuscript's publication in *The-Cryosphere*.

**General Comments**:
- **Ice sheet model spin-up**: In the manuscript, you describe a 500 year spinup of the ice sheet model so that it relaxes to a pseudo steady state. I wonder about the implications of running the model to steady state given that you are trying to match observed patterns of grounding line retreat. If Vanderford Glacier was not in steady state prior to the 18.6 km grounding line retreat, is this spinup to steady state really necessary (or appropriate)? Have you checked that the rate of ice mass change of Vanderford Glacier at the end of your spin-up is in somewhat of agreement with observations? How about the simulated velocities? You could test the impact of this steady state by perhaps performing an additional experiment that starts directly after the 2 year relaxation period and comparing the results to the corresponding experiment that started from steady state. Overall, I think that some justification of this spinup, discussion of its impacts on your results (possibly highlighted as an uncertainty in the later stages of the Discussion section), and a comparison of your steady-state model to present day observations (velocity and mass loss) would be very helpful.
- **Reversibility of retreat**: I was happy (possibly relieved) to see that in all of your experiments, simulated retreat and the grounding line flux stabilized back to conditions at the start of the simulation once the forcing was reverted back to what was applied in the spinup. I think that this is an important result and is worthy of being highlighted in the text! In your simulations, you show that retreat of Vanderford Glacier is not irreversible and that, in its current configuration, Vanderford is not undergoing MISI.

- **Figure references**: I noticed a couple of incorrect figure references. I tried to catch as many of them as possible in the line comments below, but also wanted to highlight it here in case I missed any. It would be great to double check these.

**Line Comments**:
- L57-60: How do these satellite estimates perform over heavily crevassed ice shelves? Would this be a factor as well? Also, what are their resolutions?
- L78-80: This might be a good place to point to figure-1, which shows the model domain.
- L146: In your steady-state runs, do you also check that mass loss is steady as well? Is the steady state the same for all friction laws? How does the steady state geometry and ice velocity compare to the initial model state? Have you tested the impact of starting your perturbation experiments directly after the 2-year relaxation period (since Vanderford likely has not been in steady state). I'm wondering if this choice to spin up your model for such a long time feeds back on your results in some way?
- L150: Change "simulate" to "perform"
- L151: Perhaps it might be clearer to say "we simulate a series of perturbation experiments for each of the three friction laws that run for 100 years."
- If the model is spun-up for 500 years, why do you need to further spin up each experiment for another 5 years?
- L158-159: What does it mean to remove the response of the control experiment from each perturbation experiment? Can you be more specific here? Is this specific to the grounding line flux, or does this also include the response of the grounding line as well?
- L230: Does this point to the fact that MISI is not in play here and that retreat of Vanderford is reversible if forcing decreases?
- Sec. 3.1: Also, do you know what caused this large jump in grounding line retreat in the Weertman $M_{Davidson}$ simulation (yellow dotted line in figure 6b)? It is odd that you see this spike in grounding line retreat, but do not see a corresponding jump in Delta-VAF or Delta-ice volume.
- L242: Figure pointer to 5c seems incorrect(this figure shows grounding line retreat in the linear calving experiments). Did you mean 5b? Same with pointer on L271 to fig. 7j (should this be 7i?)
- L286: Fig. 7j references the basal melt perturbation experiments, do you mean fig. 7l? Same for figure reference in L294.
- L363: In addition to in situ measurements, what about the opportunity to back out high resolution ice shelf melt rate maps using techniques such as the one described in Zinck et al. (2023); https://tc.copernicus.org/articles/17/3785/2023/? Given that direct observations of ice shelf melt are difficult/costly to obtain, it might be worth discussing other avenues.

**Figure Comments**:
- Fig. 2: In the caption, you say "current mass loss estimates for Vincennes Bay", but is this accurate since you are showing ice shelf melt rates and calving rates? Might consider removing this qualifying sentence.
- Fig. 3: It is difficult to tell which experiments are bolded (I did not realize any were bold until I read the caption). Can you make this stick out more?

- Fig. 6: Is panel-b showing the location of the grounding line along the flowline (e.g., in the Weertman M_{Davidson} experiment, the grounding line was ~2.5 km retreated inland from year 20-25, but then advanced back to its present day position at year 30), or is this the retreat rate? It might be nice to briefly mention this in the figure caption or corresponding section of the paper.
- Fig. 7: In the caption, I am a bit confused at your description of what the purple line is in panels d-f. Is the purple line the location of the 2020 grounding line along the flowline?
- Fig. 10: The orange line is a bit difficult to see given the colormap used. Can you try to make these stand out a bit more?

---

## Author Comment (AC1)

**Assessing the sensitivity of the Vanderford Glacier, East Antarctica, to basal melt and calving**

**Author Responses to Referee Comments**

Lawrence A. Bird      Felicity S. McCormack      Johanna Beckmann

Richard S. Jones      Andrew N. Mackintosh

Dear Cheng Gong:

We thank both reviewers for their comprehensive review of our manuscript and the improvements their constructive comments will bring to the manuscript. Here, we respond to comments from Benjamin Getraer (Reviewer 1). Below, we respond (in blue) to each of their comments (in black). We respond to comments from Tyler Pelle (Reviewer 2) in a separate document.

Sincerely,

Lawrence Bird and co-authors

**General comments**

This manuscript contains valuable information and impressively comprehensive experimental results for guiding future modeling, data collection, and interpretation of Vanderford Glacier. However, currently the takeaways are not sufficiently clear, certain aspects of the methodology need to be more clearly explained and defended, and in general the manuscript needs to be edited for clarity and some technical errors.

The results here substantially contribute to the general understanding of Vanderford Glacier retreat, showing that observed calving has occurred in passive, non-buttressing ice, and did not contribute to the observed retreat rates. The authors' conclusion that basal melting likely is responsible for the observed grounding line retreat at Vanderford Glacier is important and well supported by their sensitivity testing, which suggests that high but plausible basal melt rates can reproduce observed retreat.

The authors interpret that because altimetry based estimates of melt rates are too low to reproduce observed grounding line retreat in their model, these observational datasets may be incorrect in this region. This conclusion may be valid but ought to be supported and discussed more quantitatively—two ways this analysis could be further improved are suggested in Specific Comments.

As the manuscript discusses in a few places, there are other factors which play a significant role in the simulated grounding line dynamics, and additional observational constraints which would improve future modeling efforts, beyond the melt rate itself. The presentation of these other factors can be improved, as currently the emphasis on the melt rate as a source of error and uncertainty in the abstract and conclusion does not reflect the more balanced discussion section. I offer more specific comments on balancing the discussion of sources of error in Specific Comments. Some of the scientific methods and assumptions need to be more clearly outlined and/or defended. These are also addressed in Specific Comments.

*We are glad that the reviewer feels the manuscript contributes substantially to the general understanding of Vanderford Glacier and we thank them for their constructive feedback.*

*Both reviewers raise some good points around the clarity and cohesion between the methodology, discussion, and implications. We believe this may stem from the fact that in the original manuscript, the aims were not expressed clearly. As such, we have rewritten the aims to be more precise/correct, as follows (Lines 66-70):*

*"The aim of this study is to assess the sensitivity of mass loss and grounding line retreat at Vanderford Glacier to sub-ice shelf basal melt and calving. We use time-evolving numerical ice sheet model simulations to address the following research questions: 1) can satellite-derived estimates of basal melt and ice-front retreat generate the magnitude of observed grounding line retreat at Vanderford Glacier between 1996-2020; and, if not, 2) what magnitude of basal melt and/or calving is required to generate grounding line retreat of a similar magnitude to observed?*

*Although the aim of this study is not to directly replicate recent trends at Vanderford Glacier, by addressing the above questions, we are able to infer the likely driver(s) of recent historical changes in mass loss and grounding line retreat at Vanderford Glacier."*

*The rewording of the study aim above better reflects the original intention of the study, why the methodology was developed as it was, and is consistent with the findings presented in the manuscript. In light of this rewording, we have also adjusted the order of the first two paragraphs presented in the discussion for improved and consistent structure/readability.*

**Specific Comments**

**Abstract**

As written, the abstract suggests that both observed melt rates and observed calving rates are insufficient to drive observed grounding line retreat (lines 8–11), and then goes on to state that the grounding line retreat was likely driven by melting and not calving (lines 11–13). This structuring is confusing, and I think do not clearly communicate the most important findings of the paper. The manuscript suggests/concludes later in the text (lines 421–426) that it is physically impossible for the observed calving to exert any significant effect on the grounding line, which is supported by their perturbation experiments and by their calculation of the maximum buttressing field. I think this is a much more clear finding by which to reject the influence of ice calving on the observed grounding line retreat, as it stands for itself and does not require further interpretation. In contrast, the statement in the abstract that "... calving experiments suggest that $> 80\%$ ice-front retreat—well in excess of the observed ice-front retreat since 1996—needs to occur to generate grounding line retreat similar to observations," still left me unsure whether the observed ice front retreat could account for some of the grounding line or not. This statement also sets up a dichotomy with the previous parallel statement on melt rates, which creates the impression that they may be equally insufficient to drive the observed retreat. In this context, the concluding statement (lines 12–13), that "retreat... is likely to be dominated by basal melt, with an almost negligible contribution from calving," appears arbitrary and unsubstantiated. A re-framing of the abstract in a way that more clearly expresses the findings which support its conclusions would function as more accurate and less confusing summary of the paper.

We agree that a slight reframing of the abstract would help to better summarise the paper, also due to the rewording of the aim (as above). We have revised the abstract to address the reviewer's comments as follows:

*Vanderford Glacier is the fastest retreating glacier in East Antarctica; however, the driver of observed grounding line retreat remains unknown. The presence of warm modified Circumpolar Deep Water offshore Vanderford Glacier suggests that grounding line retreat may be driven by sub-ice shelf basal melt, similar to the neighbouring Totten Glacier. We use an ice sheet model to assess the sensitivity of mass loss and grounding line retreat at Vanderford Glacier to sub-ice shelf basal melt and ice-front retreat. We compare simulations forced by satellite-derived estimates of long-term mean annual basal melt and observed annual ice-front retreat, and varying magnitude idealised basal melt and ice-front retreat. Forcing the model with satellite-derived basal melt rates and observed ice-front retreat results in minimal grounding line retreat, suggesting that these forcings cannot generate grounding line retreat of a similar magnitude to observations, and that observed ice-front retreat has removed only passive ice from the ice-shelf. In our sensitivity experiments, ice-front retreat >80% of the ice shelf length fails to produce grounding line retreat of a similar magnitude to observations. Instead, basal melt rates >50 m yr$^{-1}$ at the grounding line - more than twice current estimates - are needed. Our results*

*suggest that grounding line retreat and dynamic mass loss at Vanderford Glacier is likely to be dominated by basal melting higher than current satellite-derived estimates, highlighting the need for improved constraints on basal melt estimates in the Vincennes Bay region, and that ice-front retreat has an almost negligible impact on sustained grounding line retreat.*

**Interpretation of "low" melt rate observations**

The authors suggest that observational datasets of melt rates calculated from satellite altimetry may be incorrect in this region. Two suggestions are made to address and support this claim:

1. The primary cause of uncertainty in melt rate estimations that is cited by the authors is that portions of the ice shelf may be floating non-hydrostatically. This source of bias and uncertainty has been independently estimated in the Vincennes Bay region by Chartrand and Howat (2023) (whom the authors cite) at $-0.8 \pm 12.7$ m w.e. a$-1$ (mean and standard of deviation, which the authors do not directly cite). Whether the further end of this range, which suggests that melt could be underestimated by up to 13.5 m w.e. a$-1$ (or 14.7 m a$-1$ in terms of ice thickness), is sufficient to bridge the gap their model reveals between observed melt rates and grounding line retreat could be addressed directly by the authors.

We thank the reviewer for their suggestion to account for estimated uncertainties when considering current estimates of basal melt. We have included these uncertainty estimates in the discussion and added a comment to highlight that current estimates of basal melt (i.e., historical satellite-derived estimates) are always lower than the 50 m yr$^{-1}$ required in our sensitivity analyses to produce grounding line retreat of a similar magnitude to observed, even when accounting for the uncertainties in the observed melt rates. Amended text below begins on Line 361:

*"… These hydrostatic thickness uncertainties result in large uncertainties in basal mass balance estimates (-0.8 $\pm$ 12.7 m w.e. yr$^{-1}$) across Vincennes Bay, influenced largely by spatial changes in the strain rate across the ice shelves (Chartrand and Howat, 2023). We note that when accounting for the upper range of basal mass balance uncertainty, satellite-derived estimates of basal melt rates across the Vanderford ice shelf (Fig. 2) are always below 50 m yr$^{-1}$. This suggests that uncertainties in hydrostatic thickness alone cannot explain the elevated basal melt required to drive grounding line retreat at Vanderford Glacier in our sensitivity analyses. This highlights the need for improved…"*

2. In Figure 4a–b, the authors present the "Mean annual basal melt rate derived from satellite altimetry" from Paolo et al. (2022) and Davison et al. (2023), respectively. Given that these figures show dramatically different spatial patterns of basal melt/accumulation rates, a quantified measure of uncertainty in altimetry-observed melt rates could be estimated by calculating the variance in melt rates between these datasets which have 20 years of overlapping data. Comparing the variance in melt rate estimates between

these two parallel datasets with their results could better inform their discussion of observational uncertainty, and how observed melt rates compare to their inference of very high melt rates at depth.

Figure 2 shows the mean annual basal melt rate derived from satellite altimetry from Paolo et al. (2022) and Davison et al. (2023). These 2D fields show the "long-term" mean annual basal melt rates from Paolo et al. (2022) and Davison et al. (2023). That is the mean annual melt across the entire period of each available dataset (Lines 166-167). While Paolo et al. (2022) provide a time series of mean annual melt rates, the same temporally-varying data is not published by Davison et al. (2023). Since comparable time series datasets are not readily available, the only way we can compare these two datasets is via the long-term mean annual melt rates; therefore, we preferenced the suggestion above (Line 105 of this document) to comment on uncertainty within these basal melt estimates.

**Other sources of uncertainty**

An important finding of this manuscript is that current estimates of melt rates from the observational record are insufficient to reproduce the observed extent of grounding line retreat in the numerical simulations presented here.

We have reworded the aim of the study (please see the response to the the reviewer's general comments) to highlight that the focus is to investigate the sensitivity of the Vanderford Glacier to sub-ice shelf melt and calving to determine the likely driver of observed change, rather than to match observed patterns of grounding line retreat.

Extensive testing of potential sources of error in the model which could yield these results is out of the scope of the experiments which this manuscript presents, but the treatment of sources of error ought to be presented evenly in the abstract and conclusion. The authors highlight that bed topography has uncertainties which could exceed 500 meters in the region upstream of the grounding line. The observed melt rates are introduced as perturbations to a steady state spin up which may not share a similar geometry to the real historical ice sheet. If there is reason to conclude that uncertainty in melt rates could be a more important source of error than uncertainty in bed topography, or initial ice sheet state, in the grounding line dynamics of this model of Vanderford, then there should be clear evidence presented supporting that argument.

This is a great point. We agree that uncertainties associated with bed topography may have important implications for the perturbation experiments performed here. For instance, if the bed topography were deeper than current estimates, particularly in the Vanderford Trench upstream of the present-day grounding line (where uncertainties are > 500 m in some places due to insufficient radar coverage), additional grounding line retreat under lower magnitude melt rates than simulated in our experiments may be possible. This is an important point to consider and one that we discuss on Lines 389-391, considering why relatively high melt rates were required to generate grounding line retreat of a similar magnitude to that observed.

Unfortunately, considering the sensitivity of the system to uncertainties in bed topography is outside of the scope of this study – which specifically intends to understand the role of different forcings in inciting grounding line retreat at Vanderford Glacier – but this question certainly warrants further investigation (e.g., as in Castleman et al., 2022).

By initial ice sheet state, we assume the reviewer is referring to the steady-state after the spin-up. We chose this approach as it removes any inertia inherent in the system, so that we can test the response to perturbations, as per our aim (please see the response to the reviewer's general comments). We further address the implications of the model initialisation in response to a later comment from the reviewer (Line 253 of this document).

Additionally, while the authors suggest in-situ measurement of melt rates is an important future step to take (lines 363 and 484), this suggestion is not clearly supported by their results or discussion. The expense, difficulty, and lack of spatial and temporal coverage associated with direct in-situ measurements of basal melt rates make it unclear how such measurements would actually be incorporated into their model or contextualized. Simply put, a few measurements in a vast system that are very high or very low or in between do not necessarily help to characterize the behavior of the system at a useful scale. In comparison, other suggestions put forward in the discussion section are much better supported by the results and discussion of the manuscript: improved estimates of ice shelf geometry to constrain the existing, ice-shelf wide, remote sensed data which are already usable in the existing model; improved bathymetry and ocean state measurements around the ice shelf to constrain ocean circulation models and estimates of heat flux into the ice shelf cavity; improved bed topography beneath grounded and floating ice to constrain the ice sheet geometry and grounding line dynamics.

We appreciate the reviewer's comment that in-situ melt estimates are challenging, although we acknowledge previous studies that demonstrate that even a few, or pointwise, measurements can provide tremendous insight into melt and melt processes (e.g. Vaňková et al., 2023; Davis et al., 2023).

As pointed out by the reviewer, there are possibly some smaller steps that can be taken to make tangible gains in our understanding of basal melting in this region. The most obvious step is to improve the bathymetry, so that ocean model estimates of melt are more accurate. This is the current focus of a separate study. The kinds of measurements highlighted in the discussion – ApRES, oceanographic estimates – are incredibly valuable sources of information to constrain such ocean model estimates. Furthermore, the ice-ocean community recognises the immense challenges in reconciling observations of melt, and how it is represented in both ocean and ice sheet models, as discussed in McCormack et al. (2024). Indeed, the Joint Commission on Ice-Ocean Interactions is currently developing a framework that provides guidance on how to reconcile melt rates from different sources (geophysics, oceanography, and satellite), and how to integrate them into models in a useful and appropriate way (see FUSION in McCormack et al., 2024).

While satellite-derived estimates provide a useful method to infer basal melt rates, they rely on the assumption of hydrostatic equilibrium. As we discuss (Lines 354-364), the validity of

this assumption is challenged in small dynamics systems in Antarctica, such as Vincennes Bay, where steep ice thickness gradients exist, and on Vanderford Glacier ice shelf in particular, which is heavily crevassed. We have updated Line 362 as follows:

*"…This highlights the need for improved estimates of basal melt across Vincennes Bay ice shelves. While recent advances in satellite-derived basal melt estimates (e.g. Shean et al., 2019; Paolo et al., 2022; Davison et al., 2023; Zinck et al., 2023) allow for high-resolution estimates of sub-ice shelf basal melt, some simplifying assumptions (e.g. hydrostatic equilibrium) have limited applicability in small dynamic systems such as Vincennes Bay, which highlights the ongoing need for direct observations (e.g., geophysical or oceanographic) to help constrain remotely-sensed estimates (McCormack et al., 2024)."*

**Scientific Methods and Assumptions**

1. Lowering and smoothing of the bed underneath the ice shelf (lines 86–91). A more clear explanation of what happens in the models if these adjustments are not made would better allow the reader to judge the validity and possible bias introduced by this adjustment. The statement that "this approach may limit any grounding line advance across model simulations" leaves unanswered the question of what effect such a bias could have. If it does prevent grounding line advance beyond the initially imposed geometry, it may obscure a tendency of the model to advance in the spin-up stage.

Mapped multibeam bathymetry offshore the Vanderford Glacier shows that there are large errors in existing bathymetry estimates (e.g. BedMachine v3/IBCSO v2) in this region (Figure 1). Given a mean difference of 840 m between BedMachine v3 bathymetry and mapped bathymetry in coincident locations at the ice-front, it is reasonable to conclude that the BedMachine estimate of bathymetry here has some limitations, and hence that some adjustment to the sub-ice shelf bathymetry is reasonable. Since BedMachine v3 enforces a minimum water cavity depth of 1 m below the ice shelf base (Morlighem et al., 2020), sub-ice shelf bathymetry below the Vanderford Glacier from BedMachine v3 includes localised regions of steep gradients and likely unrealistic topography. To prevent buttressing forces acting on the ice shelf as ice advects through the ice shelf, we smooth the bed to remove these localised steep gradients (see Figure 1 below). During the spin-up period, the grounding line shows a tendency towards localised retreat rather than advance, suggesting that the adjustment to the sub-ice bathymetry does not bias the model spin-up by preventing grounding line advance. Furthermore, as described on Lines 92-95, since Vanderford Glacier has experienced retreat in recent decades and the initial grounding line represents the most advanced grounding line since observations began, we do not expect the grounding line to advance beyond this position. Additional model experiments that assess the impact of the model spin-up (Figure 2) also suggest a tendency for grounding line retreat and not advance.

We have discussed this on Line 88 as follows:

*"...deep bathymetry offshore the Vanderford ice-front. Adjustment of the sub-ice shelf bathymetry ensures that the ice shelf does not re-ground on unrealistic high points in the ice shelf cavity, introducing buttressing forces. We enforce a minimum water depth..."*

[Figure]

Figure 1: Mapped bathymetry discrepancy and adjusted bathymetry. (a) BedMachine v3 (Morlighem et al., 2020) bed topography. Elevation difference between BedMachine v3 and mapped multibeam bathymetry from Commonwealth of Australia (2022). (b) Bed topography included in our model set-up. Red line denotes the model domain boundary. Blue line denotes the initial grounding line (MEaSUREs v2).

2. The use of a 500 year spin-up steady state from which the perturbations are conducted could be introduced and defended much more clearly. My understanding from reading the manuscript is that there is an implicit assumption that the forcings which caused grounding line retreat over the observational record ought to be able to cause similar grounding line retreat when introduced as a perturbation from the 500 year steady state model. If this interpretation is correct, it should be much more clearly stated in the manuscript. Importantly, the state of the ice domain (i.e. thickness, velocity, strain rates) after the 500 year spin-up in comparison to the initialization fields from observations should be shown somewhere in the main text or in the supplement. Any significant differences between the model state before and after the spin up should be discussed in the manuscript as to how those differences should be interpreted, and how they may impact the results.

We thank the reviewer for raising this point and agree that this is an important factor to explicitly comment on and that was missing in the original manuscript. The focus of our study is to conduct a sensitivity analysis of different mass loss mechanisms on Vanderford Glacier.

Hence, we want to ensure that the system responds *only* to the perturbation applied to the different forcings and is not influenced by inherent trends associated with the model initialisation procedure. For this reason, we initialise the model to a pseudo-steady state using the 500-year spin-up (lines 143-147). We also simulate a *Ctrl* experiment using forcings consistent with the spin-up (lines 157-158), and in our analysis (Section 3), we present results with respect to the control (line 158-159), i.e., subtracting the control from the perturbation simulations, to remove any model drift, or influences on ice thickness or ice velocity that do not arise from our basal melt or calving perturbations. Present-day conditions at Vanderford Glacier are likely not in a steady-state; however, initialising our model to present-day conditions (e.g. with the observed trend in thickness over the historical period) would limit our ability to untangle the changes arising from basal melt and calving, and those that arise due to inertia in the system.

We have completed some additional model runs to assess the impact of our model initialisation choice on select basal melt perturbation experiments and discuss these below. Figure 2 displays results of additional model runs completed for $M_{10}$ - $M_{50}$ for each friction law, without the 500-year spin-up step. That is, we apply the perturbation period immediately after the initial 2-year relaxation step. We present these additional model runs in the first column and the corresponding model runs which use the 500-year spin-up step in the second column. Note that we have adjusted the presentation of grounding line retreat based on a comment from Tyler Pelle (Reviewer 2).

Figure 2 shows that the relative change of $GL_{flux}$, grounding line position, $VAF$, and Ice volume, compared to a control experiment, does not differ considerably from our simulations that use a spin-up to pseudo-steady state. The largest differences are in the: 1) time it takes for the grounding line to re-advance to its original location, and 2) variability between the different friction laws. Figure 2 c-d show that with no spin-up, the grounding line takes longer to return to its initial location for some experiments than when the perturbation is applied to a system in pseudo-steady state. This result is expected since there is likely inherent inertia within the present-day system towards grounding line retreat that arises from the system response to previous basal melting and/or calving. The response to different friction laws shows a slightly different pattern than our original experiments, with Weertman experiments consistently generating the smallest change in $VAF$ compared to other friction laws; however, all experiments yield the same pattern and magnitude of change as our original experiments.

Based on the results of these additional experiments, it is clear that the initialisation and spin-up procedure is appropriate in isolating the instantaneous effects of sub-ice shelf basal melt and ice-front retreat because they ensure that other trends in the system response to past forcings are removed. Given the aims of our study and that our sensitivity analysis uses a model configuration in pseudo-steady state, it is more appropriate to present the change in key variables (e.g. ice thickness, ice velocity, and grounding line flux) over the course of the spin-up period, rather than to compare the final model fields to observations. We have added a Figure to the Supplementary Information to show the evolution of these variables throughout the spin-up period.

We have made the following in-text adjustments to comment on our choice of model initialisation and to point to the additional figure in the Supplementary Information:

Line 146: *"At the end of this 500-year simulation, ice velocities and geometry are in pseudo-steady state (Fig. Sx), and these fields are used as the initial conditions for all perturbation experiments (Sect 2.3)."*

Line 464 (added): *"Our choice of model initialisation and spin-up to pseudo-steady state directly addresses the study aim, and ensures that the system responds only to the instantaneous perturbation applied to the different forcings and is not influenced by any inertia or trends within the system. This approach allows us to untangle the changes in grounding line flux, grounding line migration, volume above floatation, and ice volume that arise from basal melt and ice-front retreat, independently and in combination. Importantly, this approach limits our ability to comment on the current state of Vanderford Glacier (i.e. whether or not it may be undergoing irreversible grounding line retreat). To accurately assess the current state, or to comment on potential future behaviours at Vanderford Glacier, additional modelling simulations which use a present-day model initialisation (i.e. to accurately match recent trends in mass loss and spatial patterns of ice thickness, ice velocity, and grounding line position) are recommended. Simulations of present-day behaviour require improved observations of basal melt, particularly close to the grounding line, and a more accurate representation of bathymetry in ocean models used to parametrise sub-ice shelf basal melt is needed to support simulations of future behaviour."*

**Other comments**

**line 8** "… instead, basal melt rates in excess of 50 m $yr^{-1}$ at the grounding line…" It is unclear how much more 50 m $yr^{-1}$ is than the observed rates, making this statement vague and hard to interpret.

We have edited the abstract for clarity as per a previous comment (Line 60 of this document) – please see adjustments to the abstract at the top of this document.

**lines 49–50** The phrasing of this idea implies that we should not expect high local variability in melt rates, even though that is quite common. I agree that high melt rates nearby, forced by similar ocean water, supports the plausibility of similarly high melt rates under Vanderford, but only in the context of other evidence. The disparity between observed Totten and Vanderford melt rates alone, without the context of these results, is not necessarily surprising at all. This idea might be better suited for the discussion section.

We chose to leave this information here to highlight the fact that despite similar mCDW temperatures in Vincennes Bay and at Totten Glacier (Lines 44-45), there are notable differences in the basal melt rates. We agree that while localised variations in melt rates are not uncommon, evidence of similar regional ocean conditions might suggest that basal melt rates

[Figure]

Figure 2: Comparison of perturbation experiments with different initialisation/spin-up conditions. Experiments without a 500-year spin-up to pseudo-steady state are presented in the first column. Experiments with a 500-year spin-up to pseudo-steady state are presented in the second column. Relative change (compared to the *Ctrl* experiment for each friction law) of: (a-b) Grounding line flux, (c-d) Grounding line position calculated along the central flowline show in Fig. 2c. Negative numbers represent grounding line retreat and positive numbers represent grounding line advance, relative to the Ctrl experiment. (e-f) Volume above floatation, (g-h) Ice volume. Grey shaded area denotes the perturbation period.

would be more similar in these neighbouring systems. This information provides context and motivation for the study.

**lines 55–60** These sentences are not clearly connected to the introduction and seem at least partially related to or repeated in the discussion.

We chose to leave this information here as it provides context and motivation for the study. While previous studies have begun to explore contributions from basal melt and calving to Antarctic mass loss based on satellite observations, the accuracy of basal melt predictions remains a critical limitation in our ability to correctly attribute different mass loss contributions from different processes around Antarctica.

**Figure 2** The Totten insets are blocking the Underwood ice shelf which is otherwise visible in the other figures. Because the colorbar is scaled to the higher melt rates on Totten, it is hard to actually interpret the range of observed melt rates on Vanderford. Contours, labels, or different color limits could be better. Additionally, this figure does not make clear what the spatial extent of the observed melt rates are: does the observed melt rate field evolve in time with the ice shelf cavity to follow the grounding line retreat? Or are they only provided for this spatial extent (of the unretreated grounding line)?

Basal melt estimates are not provided for Underwood Glacier by Paolo et al. (2022) and Davison et al. (2023), so we use this region to display the estimates from Totten Glacier. We use the same colour scale for both Vanderford and Totten to clearly highlight the difference in the magnitude of the basal melt rates between the two systems. The spatial extent of the data is consistent with the spatial extent of the data provided by Paolo et al. (2022) and Davison et al. (2023). For $M_{Paolo}$ and $M_{Davison}$ experiments, we interpolate these fields across the model mesh to ensure that basal melt rates are appropriately provided for all model nodes as the grounding line retreats.

We have updated the figure to include a spatial scale for the Totten Glacier inset and have updated the caption to explicitly state how melt rates are applied to the model domain, as follows:

*"…The extent of the Totten Glacier inset in (a) and (b) is shown in Fig 1a. No basal melt estimates are available for Underwood Glacier. For $M_{Paolo}$ and $M_{Davison}$ experiments, we interpolate these basal melt fields across the model mesh to ensure that basal melt rates are provided appropriately as grounding line migration occurs."*

**line 69–70** "…can current estimates of basal melt and ice-front retreat…." This sentence confused me because it sounded like the experiments would only use observed melt from a single snapshot in time (current estimates). The observed melt rate datasets and ice front retreats span at least 25 years of the observational record.

We have reworded the aim of the study and the associated research questions (please see the response to the reviewer's general comments) to provide clarity on how observational datasets are used.

**line 106–109** The quantities here should be given units similar to lines 111–112.

We have added units to $\tau_b$, $C_w$, $C_b$, $C_s$, $u_b$, and $N$, as follows:

*"...where $\tau_b$ (Pa) is basal shear stress, $C_w$ (kg$^{1/2}$ m$^{-2/3}$ s$^{-5/6}$), $C_b$ (s$^{1/2}$ m$^{-1/2}$), and $C_s$ (kg$^{1/2}$ m$^{-2/3}$ s$^{-5/6}$) are friction coefficients for the Weertman, Budd, and Schoof friction laws, respectively, $u_b$ (m s$^{-1}$) is the basal velocity, m is a positive exponent set to 1/3, and $C_{max}$ is Iken's bound, set to 0.5 (Brondex et al., 2017). In Eq.2 and Eq.3, N (Pa) relates to the effective pressure..."*

**line 166** "...mean annual basal melt rates...." This seems to refer to an average across each year resulting in a different "two-dimensional mean annual melt rate field" (line 145) for each year. If that is the case, that should be clarified. Similarly, clarification is needed for the "Mean annual basal melt rate" fields shown in Figure 2a–b. My assumption in Figure 2 was that this is a two-dimensional field which forms the average per-year melt rate across the entire observational record (i.e. not just a single year). More explanation is needed to interpret what the mean annual basal melt rate fields actually are, and whether they are averaged across the whole record or if there is a different field for each year.

We use the term "mean annual basal melt rate" to refer to the long-term mean annual basal melt rates as calculated and published by Paolo et al. (2022) and Davison et al. (2023). These represent the long-term mean annual melt rate across the period of record for each dataset (Line 167). The mean annual melt rate field is held constant throughout the perturbation period (Line 170-171).

For clarity, we updated line 166 as follows:

*"...we use long-term mean annual basal melt rates (i.e. 2-dimensional fields) derived from... "*

And update the caption for Figure 2 to use the term *"long-term mean annual basal melt rate"*.

**Figure 5** Elevation shown is from BedMachine—is this showing the elevation with the lowering and smoothing of the bed? It would be preferable to see the bed elevation as it is actually implemented in the model.

We have updated Figure 5 to include the bed topography used in the model which includes the adjustment of the sub-ice shelf bathymetry.

**lines 217–218** How different is the flux across the grounding line in the spun up steady state from the actual observations used to initialize the model?

We agree that it is important to consider the evolution of the system throughout the spin-up period. We have added a Figure to the Supplementary Information to show the evolution of ice thickness, ice velocity, and grounding line flux throughout the spin-up period (line 146). In summary, all fields display the largest changes over the first ~50 years before stabilising throughout the remainder of the spin-up period. The evolution throughout the spin-up period is similar across all friction laws.

**Figure 6b** Why does the $M_{Davison}$–Weertman grounding line retreat so far and so suddenly? This retreat does not seem to be reflected in $\Delta$VAF (Figure 6c) or in the map (Figure 5a).

Figure 3 shows the grounding line position at the beginning and end of the perturbation period (5 and 30 years, respectively), and the end of the simulation (100 years). Figure 3 also shows the change in ice thickness between the end of the perturbation period and the end of the simulation. A small change in ice thickness (i.e. $< 30$ m) in a localised region along the flowline used to calculate grounding line retreat suggests that the rapid retreat observed in the Weertman $M_{Davison}$ experiment is due to minimal ungrounding of a localised area and is not indicative of notable or widespread grounding line retreat. Since this region is so small and only experiences minimal thinning, this signal is not observed in $\Delta$ VAF or $\Delta$ Ice volume. Furthermore, due to only minimal thinning, ice re-grounds rapidly once the perturbation is removed, explaining the rapid re-advance of the grounding line.

[Figure]

Figure 3: Weertman $M_{Davison}$ grounding lines and thickness change. Thickness change is shown as the difference between ice thickness at the end of the simulation period (T = 100 years) and at the end of the perturbation period (T = 30 years) when the grounding line is at its most retreated. Purple line denotes the flowline along which grounding line retreat is calculated.

**lines 339–340** The melt rates required to drive grounding line retreat in the model are twice those of the observational data: This could be a clarifying result to add to the abstract, as it

contextualizes what 50 m yr$^{-1}$ means in relation to observations.

We have edited the abstract for clarity as per a previous comment (Line 60 of this document) – please see adjustments to the abstract at the top of this document.

**line 344** This struck me as a key finding: if removing the entire ice shelf relatively suddenly is actually not nearly enough to drive the rate of retreat observed, then it seems that melt rates needed to remain high at depth close to the grounding line as the ice cavity evolved.

This is a great point. We have edited the abstract for clarity as per a previous comment (Line 60 of this document) – please see adjustments to the abstract at the top of this document. We also discuss the role of melt distribution across the ice-shelf on Lines 403-410 and note that there remains debate within the community about the importance of melt at the grounding line compared to shelf-wide melt.

**lines 417–420** Mode-3 melting is associated with melting close to the ice fronts, generally in passive zones which contribute little to buttressing (Adusumilli et al., 2020). In this model, removing the front portions of the ice shelf entirely does almost nothing. Why would this type of melting be worse than melting at depth dominated by CDW?

This is a good point that we have clarified in the updated manuscript. In particular, a reduction in sea ice cover in this region is expected to result in a regime shift away from mCDW dominated (mode 2) melting towards surface dominated (mode 3) melting (Naughten et al., 2018). Our experiments using a constant melt rate across the entire ice shelf show large portions of the ice shelf removed, beyond regions of passive ice. These experiments suggest that elevated melting across shallow ice (i.e. compared to $M_s$ used in basal melt experiments) may have implications for buttressing in the future. We have modified text on Lines 417-420 as follows:

*"…a regime shift towards mode 3-dominated melt could increase mass loss from Vanderford Glacier as large portions of the ice shelf are removed with elevated melting across shallow ice regions (Fig. S6). More widespread mass loss from the ice shelf, beyond the region of passive ice, may lead to reduced buttressing forces and accelerated grounding line retreat."*

**line 432** "Our simulations show that the Vanderford Glacier system responds rapidly to calving events." This was confusing to me, as Figure 7 shows almost no response to most calving, and if anything a delayed, ongoing response to large calving.

We thank the reviewer for this comment and agree that the wording here could have been clearer in the original manuscript. Here, we comment on the response of the system as a whole (i.e. including changes to $GL_{flux}$, $VAF$, and Ice volume) and not only on the influence of calving on grounding line retreat, which, as you mention, is more delayed. We do not comment on the magnitude of the influence here, but only the timing of the response. We have modified the text on Line 432 to be clear on this as follows:

*"Our simulations show that the Vanderford Glacier system generally responds rapidly to calving events, despite a more gradual response from Cl experiments and a more delayed response*

*in grounding line retreat compared to basal melt experiments. The overall response is generally independent of the process by which calving occurs (i.e. edge-wasting or major calving events)."*

**lines 474–476** This could include the results that observed calving have had no effect. The phrasing here is slightly convoluted and confusing.

The results of the observed calving experiments are described on the preceding line (Line 472). Here, we specify that we are referring to the idealised calving experiments (line 474):

*"…Our idealised calving experiments…"*

**lines 481–488** The final paragraph of the conclusion includes vague phrasing and does not highlight a strong implication of the results. Overall the conclusion repeats a lot of the discussion. My takeaway from the manuscript overall is that there is an unresolved incompatibility between modeled dynamics, observed forcings, and observed dynamics. A clear and succinct outline of those incompatibilities, and a stance taken on what the implication of those incompatibilities are until they are resolved would be enlightening and interesting.

We agree that the conclusion should clearly present the implications of the results. Having reworded the aim of the study, we opted not to make changes to the final paragraph of the conclusion as we feel it now better aligns with the overall aim and is well supported by the findings. This is supported by comments from Reviewer 2 that the conclusions were well supported, the manuscript was well written, and that the results of the study will be important for future ice sheet modelling studies in this region. As discussed in response to another of Reviewer 1's comments, we think it is important to highlight the remaining need for additional observations to help constrain remotely-senses datasets, which we do in this final paragraph.

**Technical Corrections**

**Figure 6c–d** y-axes are labeled as changes in volumes with units of Gt. The labels should be corrected.

We chose to leave units in Gt to be consistent with common nomenclature in IPCC reports, but specify in the figure caption for Fig. 6 and Fig. 7 that $VAF$ and Ice volume are converted to ice mass in Gt, as follows:

*"Figure 6. …(c) Volume above floatation, and (d) Ice volume. Volume above floatation and Ice volume are converted to mass in Gt…."*

*"Figure 7. …(g-i) Volume above floatation, and (j-l) Ice volume. Volume above floatation and Ice volume are converted to mass in Gt…."*

**line 612** space in doi link

Corrected.

**References**

Adusumilli, S., Fricker, H. A., Medley, B., Padman, L., and Siegfried, M. R.: Interannual Variations in Meltwater Input to the Southern Ocean from Antarctic Ice Shelves, Nature Geoscience, 13, 616–620, https://doi.org/10.1038/s41561-020-0616-z, 2020.

Brondex, J., Gagliardini, O., Gillet-Chaulet, F., and Durand, G.: Sensitivity of Grounding Line Dynamics to the Choice of the Friction Law, Journal of Glaciology, 63, 854–866, https://doi.org/10.1017/jog.2017.51, 2017.

Castleman, B. A., Schlegel, N.-J., Caron, L., Larour, E., and Khazendar, A.: Derivation of Bedrock Topography Measurement Requirements for the Reduction of Uncertainty in Ice-Sheet Model Projections of Thwaites Glacier, The Cryosphere, 16, 761–778, https://doi.org/10.5194/tc-16-761-2022, 2022.

Chartrand, A. M. and Howat, I. M.: A Comparison of Contemporaneous Airborne Altimetry and Ice-Thickness Measurements of Antarctic Ice Shelves, Journal of Glaciology, pp. 1–14, https://doi.org/10.1017/jog.2023.49, 2023.

Commonwealth of Australia: RSV Nuyina Voyage 2 2021-22 Voyage Data, Southern Ocean, Antarctica, Ver. 1, https://doi.org/10.26179/ZZ6J-E834, 2022.

Davis, P. E. D., Nicholls, K. W., Holland, D. M., Schmidt, B. E., Washam, P., Riverman, K. L., Arthern, R. J., Vaňková, I., Eayrs, C., Smith, J. A., Anker, P. G. D., Mullen, A. D., Dichek, D., Lawrence, J. D., Meister, M. M., Clyne, E., Basinski-Ferris, A., Rignot, E., Queste, B. Y., Boehme, L., Heywood, K. J., Anandakrishnan, S., and Makinson, K.: Suppressed Basal Melting in the Eastern Thwaites Glacier Grounding Zone, Nature, 614, 479–485, https://doi.org/10.1038/s41586-022-05586-0, 2023.

Davison, B. J., Hogg, A. E., Gourmelen, N., Jakob, L., Wuite, J., Nagler, T., Greene, C. A., Andreasen, J., and Engdahl, M. E.: Annual Mass Budget of Antarctic Ice Shelves from 1997 to 2021, Science Advances, 9, eadi0186, https://doi.org/10.1126/sciadv.adi0186, 2023.

McCormack, F. S., Cook, S., Goldberg, D. N., Nakayama, Y., Seroussi, H., Nias, I., An, L., Slater, D., and Hattermann, T.: The Case for a Framework for UnderStanding Ice-Ocean iNteractions (FUSION) in the Antarctic-Southern Ocean System, Elementa: Science of the Anthropocene, 12, 00 036, https://doi.org/10.1525/elementa.2024.00036, 2024.

Morlighem, M., Rignot, E., Binder, T., Blankenship, D., Drews, R., Eagles, G., Eisen, O., Ferraccioli, F., Forsberg, R., Fretwell, P., Goel, V., Greenbaum, J. S., Gudmundsson, H., Guo, J., Helm, V., Hofstede, C., Howat, I., Humbert, A., Jokat, W., Karlsson, N. B., Lee, W. S., Matsuoka, K., Millan, R., Mouginot, J., Paden, J., Pattyn, F., Roberts, J., Rosier, S., Ruppel, A., Seroussi, H., Smith, E. C., Steinhage, D., Sun, B., van den Broeke, M. R., van Ommen, T. D., van Wessem, M., and Young, D. A.: Deep Glacial Troughs and Stabilizing Ridges Unveiled beneath the Margins of the Antarctic Ice Sheet, Nature Geoscience, 13, 132–137, https://doi.org/10.1038/s41561-019-0510-8, 2020.

Naughten, K. A., Meissner, K. J., Galton-Fenzi, B. K., England, M. H., Timmermann, R., and Hellmer, H. H.: Future Projections of Antarctic Ice Shelf Melting Based on CMIP5 Scenarios, Journal of Climate, 31, 5243–5261, https://doi.org/10.1175/JCLI-D-17-0854.1, 2018.

Paolo, F. S., Gardner, A., Greene, C. A., and Schlegel, N.: MEaSUREs ITS_LIVE Antarctic Ice Shelf Height Change and Basal Melt Rates, Version 1, https://doi.org/10.5067/SE3XH9RXQWAM, 2022.

Shean, D. E., Joughin, I. R., Dutrieux, P., Smith, B. E., and Berthier, E.: Ice Shelf Basal Melt Rates from a High-Resolution Digital Elevation Model (DEM) Record for Pine Island Glacier, Antarctica, The Cryosphere, 13, 2633–2656, https://doi.org/10.5194/tc-13-2633-2019, 2019.

Vaňková, I., Winberry, J. P., Cook, S., Nicholls, K. W., Greene, C. A., and Galton-Fenzi, B. K.: High Spatial Melt Rate Variability Near the Totten Glacier Grounding Zone Explained by New Bathymetry Inversion, Geophysical Research Letters, 50, e2023GL102 960, https://doi.org/10.1029/2023GL102960, 2023.

Zinck, A.-S. P., Wouters, B., Lambert, E., and Lhermitte, S.: Unveiling Spatial Variability within the Dotson Melt Channel through High-Resolution Basal Melt Rates from the Reference Elevation Model of Antarctica, The Cryosphere, 17, 3785–3801, https://doi.org/10.5194/tc-17-3785-2023, 2023.

---

## Author Comment (AC2)

**Assessing the sensitivity of the Vanderford Glacier, East Antarctica, to basal melt and calving**

**Author Responses to Referee Comments**

Lawrence A. Bird      Felicity S. McCormack      Johanna Beckmann
Richard S. Jones      Andrew N. Mackintosh

Dear Cheng Gong:

We thank both reviewers for their comprehensive review of our manuscript and the improvements their constructive comments will bring to the manuscript. Here, we respond to comments from Tyler Pelle (Reviewer 2). Below, we respond (in blue) to each of their comments (in black). We respond to comments from Benjamin Getraer (Reviewer 1) in a separate document.

Sincerely,

Lawrence Bird and co-authors

**Overview**

In this manuscript, Bird et al. present an ice sheet modeling study that investigates the role of ice shelf basal melting and ice shelf calving in driving observed retreat of Vanderford Glacier in East Antarctica. The author's primary conclusions are that ice shelf basal melting ($>50$ m/yr) is the primary forcer of this observed retreat and that currently available satellite estimates of basal melting are insufficient to drive the magnitude of retreat observed over the past ~25 years. On the other hand, ice shelf calving had minimal impact on the dynamic retreat of Vanderford Glacier. Overall, I found this paper to be a pleasure to read! The ice sheet model and experimental set-up are meticulously described and the writing is free from any grammatical errors. Furthermore, I find that the conclusions are generally well supported by the results and that the authors make sure to highlight major sources of uncertainty. I also believe that the results of the paper will be important for future ice sheet modeling studies in this region, as this paper presents important constraints on the choice of ice sheet friction law as well as which forcing mechanisms to prioritize in the modeling of Vanderford Glacier. Below,

I include suggestions that could improve the presentation and quality of the manuscript, but they are generally minor and should be relatively easy for the authors to address. Once this is done, I would be very supportive of this manuscript's publication in *The-Cryosphere*.

We are glad that the reviewer feels the paper will support future ice sheet modelling in this region and we thank them for their constructive feedback.

Both reviewers raise some good points around the clarity and cohesion between the methodology, discussion, and implications. We believe this may stem from the fact that in the original manuscript, the aims were not expressed clearly. As such, we have rewritten the aims to be more precise/correct, as follows (Lines 66-70):

*"The aim of this study is to assess the sensitivity of mass loss and grounding line retreat at Vanderford Glacier to sub-ice shelf basal melt and calving. We use time-evolving numerical ice sheet model simulations to address the following research questions: 1) can satellite-derived estimates of basal melt and ice-front retreat generate the magnitude of observed grounding line retreat at Vanderford Glacier between 1996-2020; and, if not, 2) what magnitude of basal melt and/or calving is required to generate grounding line retreat of a similar magnitude to observed?*

*Although the aim of this study is not to directly replicate recent trends at Vanderford Glacier, by addressing the above questions, we are able to infer the likely driver(s) of recent historical changes in mass loss and grounding line retreat at Vanderford Glacier."*

The rewording of the study aim above better reflects the original intention of the study, why the methodology was developed as it was, and is consistent with the findings presented in the manuscript. In light of this rewording, we have also adjusted the order of the first two paragraphs presented in the discussion for improved and consistent structure/readability.

**Specific Comments**

**Ice sheet model spin-up:** In the manuscript, you describe a 500 year spinup of the ice sheet model so that it relaxes to a pseudo steady state. I wonder about the implications of running the model to steady state given that you are trying to match observed patterns of grounding line retreat.

We thank the reviewer for raising this topic and agree that this is an important factor to explicitly comment on in the manuscript that is currently missing.

The focus of our study is to conduct a sensitivity analysis of different mass loss mechanisms on Vanderford Glacier, and not to match observed patterns of grounding line retreat (please see the reworded study aim in response to the reviewer's general comments). Therefore, we want to remove the influence of any other driver and remove any inertia that may exist within the present-day system, so a pseudo-steady state spin-up is appropriate.

A secondary aim is to assess the likely magnitude of basal melt required to generate grounding line retreat of similar magnitude to observations. Given the large uncertainties in basal melt estimates derived from satellite observations (see discussion on line 352-364), using a steady-state spin-up ensures that the influence of other mass loss drivers is removed, so that we can focus on assessing the magnitude of grounding line retreat directly in response to a given forcing (basal melt rate and ice-front retreat). We note that this approach means that we're not attempting to exactly replicate the patterns of observed retreat.

If Vanderford Glacier was not in steady state prior to the 18.6 km grounding line retreat, is this spinup to steady state really necessary (or appropriate)? Have you checked that the rate of ice mass change of Vanderford Glacier at the end of your spin-up is in somewhat of agreement with observations? How about the simulated velocities? You could test the impact of this steady state by perhaps performing an additional experiment that starts directly after the 2 year relaxation period and comparing the results to the corresponding experiment that started from steady state. Overall, I think that some justification of this spinup, discussion of its impacts on your results (possibly highlighted as an uncertainty in the later stages of the Discussion section), and a comparison of your steady-state model to present day observations (velocity and mass loss) would be very helpful.

Benjamin Getraer (Reviewer 1) had similar comments regarding the model spin-up. We provide the same response below to that provided in response to Reviewer 1.

We thank the reviewer for raising this point and agree that this is an important factor to explicitly comment on and that was missing in the original manuscript. The focus of our study is to conduct a sensitivity analysis of different mass loss mechanisms on Vanderford Glacier. Hence, we want to ensure that the system responds *only* to the perturbation applied to the different forcings and is not influenced by inherent trends associated with the model initialisation procedure. For this reason, we initialise the model to a pseudo-steady state using the 500-year spin-up (lines 143-147). We also simulate a *Ctrl* experiment using forcings consistent with the spin-up (lines 157-158), and in our analysis (Section 3), we present results with respect to the control (line 158-159), i.e., subtracting the control from the perturbation simulations, to remove any model drift, or influences on ice thickness or ice velocity that do not arise from our basal melt or calving perturbations. Present-day conditions at Vanderford Glacier are likely not in a steady-state; however, initialising our model to present-day conditions (e.g. with the observed trend in thickness over the historical period) would limit our ability to untangle the changes arising from basal melt and calving, and those that arise due to inertia in the system.

We have completed some additional model runs to assess the impact of our model initialisation choice on select basal melt perturbation experiments and discuss these below. Figure 1 displays results of additional model runs completed for $M_{10}$ - $M_{50}$ for each friction law, without the 500-year spin-up step. That is, we apply the perturbation period immediately after the initial 2-year relaxation step. We present these additional model runs in the first column and the corresponding model runs which use the 500-year spin-up step in the second column. Note

that we have adjusted the presentation of grounding line retreat based on a later comment from the reviewer (Line 291 of this document).

105  Figure 1 shows that the relative change of $GL_{flux}$, grounding line position, $VAF$, and Ice volume, compared to a control experiment, does not differ considerably from our simulations that use a spin-up to pseudo-steady state. The largest differences are in the: 1) time it takes for the grounding line to re-advance to its original location, and 2) variability between the different friction laws. Figure 1 c-d show that with no spin-up, the grounding line takes longer

110  to return to its initial location for some experiments than when the perturbation is applied to a system in pseudo-steady state. This result is expected since there is likely inherent inertia within the present-day system towards grounding line retreat that arises from the system response to previous basal melting and/or calving. The response to different friction laws shows a slightly different pattern than our original experiments, with Weertman experiments

115  consistently generating the smallest change in $VAF$ compared to other friction laws; however, all experiments yield the same pattern and magnitude of change as our original experiments.

Based on the results of these additional experiments, it is clear that the initialisation and spin-up procedure is appropriate in isolating the instantaneous effects of sub-ice shelf basal melt and ice-front retreat because they ensure that other trends in the system response to

120  past forcings are removed. Given the aims of our study and that our sensitivity analysis uses a model configuration in pseudo-steady state, it is more appropriate to present the change in key variables (e.g. ice thickness, ice velocity, and grounding line flux) over the course of the spin-up period, rather than to compare the final model fields to observations. We have added a Figure to the Supplementary Information to show the evolution of these variables throughout

125  the spin-up period.

We have made the following in-text adjustments to comment on our choice of model initialisation and to point to the additional figure in the Supplementary Information:

Line 146: *"At the end of this 500-year simulation, ice velocities and geometry are in pseudo-steady state (Fig. Sx), and these fields are used as the initial conditions for all perturbation*

130  *experiments (Sect 2.3)."*

Line 464 (added): *"Our choice of model initialisation and spin-up to pseudo-steady state directly addresses the study aim, and ensures that the system responds only to the instantaneous perturbation applied to the different forcings and is not influenced by any inertia or trends within the system. This approach allows us to untangle the changes in grounding line flux,*

135  *grounding line migration, volume above floatation, and ice volume that arise from basal melt and ice-front retreat, independently and in combination. Importantly, this approach limits our ability to comment on the current state of Vanderford Glacier (i.e. whether or not it may be undergoing irreversible grounding line retreat). To accurately assess the current state, or to comment on potential future behaviours at Vanderford Glacier, additional modelling*

140  *simulations which use a present-day model initialisation (i.e. to accurately match recent trends in mass loss and spatial patterns of ice thickness, ice velocity, and grounding line position) are recommended. Simulations of present-day behaviour require improved observations of basal*

*melt, particularly close to the grounding line, and a more accurate representation of bathymetry in ocean models used to parametrise sub-ice shelf basal melt is needed to support simulations of future behaviour."*

**Reversibility of retreat:** I was happy (possibly relieved) to see that in all of your experiments, simulated retreat and the grounding line flux stabilized back to conditions at the start of the simulation once the forcing was reverted back to what was applied in the spinup. I think that this is an important result and is worthy of being highlighted in the text! In your simulations, you show that retreat of Vanderford Glacier is not irreversible and that, in its current configuration, Vanderford is not undergoing MISI.

This is a good point, but has to be interpreted with caution. The focus of our study is to conduct a sensitivity analysis of different mass loss mechanisms on Vanderford Glacier, rather than to match observed patterns of grounding line retreat (please see the reworded study aim in response to the reviewer's general comments). As such, we cannot comment with certainty about the likelihood of irreversible retreat at Vanderford Glacier. Nevertheless, our simulations (with and without the spin-up to pseudo-steady state) suggest that the system can recover once perturbations are removed and the current grounding line may not be retreating irreversibly. Additional simulations with improved ocean forcing and bedrock topography, particularly close to and upstream of the current grounding line position, are required to accurately assess the reversibility of current grounding line retreat. For such a study, we'd recommend a different approach to the steady-state spin-up employed here; e.g., an initialisation/spin-up that embedded the historical trends into the simulated dH/dt.

**Figure references:** I noticed a couple of incorrect figure references. I tried to catch as many of them as possible in the line comments below, but also wanted to highlight it here in case I missed any. It would be great to double check these.

Thank you for highlighting these incorrect references. We have reviewed all figure references and made amendments where necessary.

**Line Comments:**

**L57-60** How do these satellite estimates perform over heavily crevassed ice shelves? Would this be a factor as well? Also, what are their resolutions?

We further discuss the limitations of satellite-based basal melt estimates in the discussion section (Line 352-364). Regions of steep ice thickness gradients are conducive to regions of crevassing or other ice damage (Chartrand and Howat, 2023). We have added explicit mention of crevassing on Line 356:

*"…In regions with steep ice thickness gradients (i.e. conducive to regions of heavy crevassing) and at…"*

[Figure]

Figure 1: Comparison of perturbation experiments with different initialisation/spin-up conditions. Experiments without a 500-year spin-up to pseudo-steady state are presented in the first column. Experiments with a 500-year spin-up to pseudo-steady state are presented in the second column. Relative change (compared to the *Ctrl* experiment for each friction law) of: (a-b) Grounding line flux, (c-d) Grounding line position calculated along the central flowline show in Fig. 2c. Negative numbers represent grounding line retreat and positive numbers represent grounding line advance, relative to the Ctrl experiment. (e-f) Volume above floatation, (g-h) Ice volume. Grey shaded area denotes the perturbation period.

We have added mention of the typical resolution of satellite-derived basal melt estimates to Line 58:

*"…basal melt estimates are derived at high resolution (i.e. 1-2 km) from satellite altimetry-based methods; however, these methods require various…"*

**L78-80** This might be a good place to point to figure-1, which shows the model domain.

We have moved the reference to Fig. 1 on Line 78 to point more directly to the model domain, rather than Vincennes Bay:

*"We use the Ice-sheet and Sea-level System Model (ISSM; Larour et al., 2012) to run transient simulations of the Vincennes Bay drainage basin. The model domain (Fig. 1) covers the Vincennes Bay drainage basin…"*

**L146** In your steady-state runs, do you also check that mass loss is steady as well?

We consider the evolution of the grounding line flux, as well as changes in ice thickness and ice velocity throughout the spin-up period. We have added a Figure to the Supplementary Information to show the evolution of ice thickness, ice velocity, and grounding line flux throughout the spin-up period (line 146).

Is the steady state the same for all friction laws? We initialise each friction law separately (line 127 and 142) and subsequently perform the 500-year spin-up separately for each friction law. We have added an explicit statement to this effect on line 143:

*"…we perform a 500-year spin-up simulation (Fig. 3) for each friction law where…"*

How does the steady state geometry and ice velocity compare to the initial model state?

We have added a Figure to the Supplementary Information to show the evolution of ice thickness, ice velocity, and grounding line flux throughout the spin-up period (line 146). In summary, all fields display the largest changes over the first ~50 years before stabilising throughout the remainder of the spin-up period. The evolution throughout the spin-up period is similar across all friction laws.

Have you tested the impact of starting your perturbation experiments directly after the 2-year relaxation period (since Vanderford likely has not been in steady state). I'm wondering if this choice to spin up your model for such a long time feeds back on your results in some way?

This is a good point. We address the implications of our model spin-up in response to a previous comment from the reviewer (Line 52 of this document). Please refer to our additional model experiments, discussion above, and additional figure added to the Supplementary Information in response to this question.

**L150** Change "simulate" to "perform"

Updated:

*"…basal melt and calving, we perform a series of perturbation experiments."*

**L151** Perhaps it might be clearer to say "we simulate a series of perturbation experiments for each of the three friction laws that run for 100 years."

We have updated Lines 150-151 as follows:

*"To assess the sensitivity of Vanderford Glacier to sub-ice shelf basal melt and calving, we simulate a series of perturbation experiments which run for 100 years, for each of the basal friction laws (Sect. 2.1)...."*

If the model is spun-up for 500 years, why do you need to further spin up each experiment for another 5 years?

We include a short perturbation-free period during all experiments primarily to ensure that the timestepping of calving and basal melt perturbations were correctly aligned and that the timing of discrete perturbations aligned with the frequency of requested model output. The choice of 5 years is arbitrary, but given that our initial conditions are from a pseudo-steady state ice sheet, this is unlikely to have any impact on the results.

**L158-159** What does it mean to remove the response of the control experiment from each perturbation experiment? Can you be more specific here? Is this specific to the grounding line flux, or does this also include the response of the grounding line as well?

For all reported timeseries of $GL_{flux}$, $VAF$, Grounding line retreat, and Ice volume, for each perturbation experiment, we remove the timeseries from the control experiment to isolate the effects of each perturbation. We have updated "Grounding line retreat" to be "$\Delta$ Grounding line position" to be consistent with other variables. The use of $\Delta$ in Fig. 6-8 represents the relative change of the perturbation experiment from the control experiment (i.e. the sole effect of the perturbation).

**L230** Does this point to the fact that MISI is not in play here and that retreat of Vanderford is reversible if forcing decreases?

This is an interesting question. We comment on the reversibility of retreat/MISI in response to an earlier question (Line 146 above) - this response is also appropriate/relevant here.

**Sec. 3.1** Also, do you know what caused this large jump in grounding line retreat in the Weertman $M_{Davidson}$ simulation (yellow dotted line in figure 6b)? It is odd that you see this spike in grounding line retreat, but do not see a corresponding jump in Delta-VAF or Delta-ice volume.

Benjamin Getraer (Reviewer 1) had the same comment. We provide the same response below to that provided to Reviewer 1.

Figure 2 shows the grounding line position at the beginning and end of the perturbation period (5 and 30 years, respectively), and the end of the simulation (100 years). Figure 2 also shows the change in ice thickness between the end of the perturbation period and the end of the simulation. A small change in ice thickness (i.e. < 30 m) in a localised region along the flowline used to calculate grounding line retreat suggests that the rapid retreat observed in

250 the Weertman $M_{Davison}$ experiment is due to minimal ungrounding of a localised area and is not indicative of notable or widespread grounding line retreat. Since this region is so small and only experiences minimal thinning, this signal is not observed in $\Delta$ VAF or $\Delta$ Ice volume. Furthermore, due to only minimal thinning, ice re-grounds rapidly once the perturbation is removed, explaining the rapid re-advance of the grounding line.

[Figure]

Figure 2: Weertman $M_{Davison}$ grounding lines and thickness change. Thickness change is shown as the difference between ice thickness at the end of the simulation period (T = 100 years) and at the end of the perturbation period (T = 30 years) when the grounding line is at its most retreated. Purple line denotes the flowline along which grounding line retreat is calculated.

255 **L242** Figure pointer to 5c seems incorrect(this figure shows grounding line retreat in the linear calving experiments). Did you mean 5b? Same with pointer on L271 to fig. 7j (should this be 7i?)

We have corrected this figure reference to be Fig. 5b and the reference on Line 271 to be Fig. 7i. We have reviewed all figure references and made amendments where necessary.

260 **L286** Fig. 7j references the basal melt perturbation experiments, do you mean fig. 7l? Same for figure reference in L294.

We have corrected both these figure references to be Fig. 7l. We have reviewed all figure

references and made amendments where necessary.

**L363** In addition to in situ measurements, what about the opportunity to back out high resolution ice shelf melt rate maps using techniques such as the one described in Zinck et al. (2023); https://tc.copernicus.org/articles/17/3785/2023/? Given that direct observations of ice shelf melt are difficult/costly to obtain, it might be worth discussing other avenues.

This is a great point and aligns with a similar comment made by Benjamin Getraer (Reviewer 1) about the appropriateness of collecting direct observations. As we discuss (Lines 354-364), satellite estimates often rely on the assumption of hydrostatic equilibrium; however, this assumption is challenged in regions such as Vincennes Bay. While satellite estimates provide a useful and convenient method to infer basal melt rates, current assumptions can limit their applicability to small dynamic systems in Antarctica. We have updated Line 362 as follows:

*"…This highlights the need for improved estimates of basal melt across Vincennes Bay ice shelves. While recent advances in satellite-derived basal melt estimates (e.g. Shean et al., 2019; Paolo et al., 2022; Davison et al., 2023; Zinck et al., 2023) allow for high-resolution estimates of sub-ice shelf basal melt, some simplifying assumptions (e.g. hydrostatic equilibrium) have limited applicability in small dynamic systems such as Vincennes Bay, which highlights the ongoing need for direct observations (e.g., geophysical or oceanographic) to help constrain remotely-sensed estimates (McCormack et al., 2024)."*

**Figure Comments:**

**Fig. 2** In the caption, you say "current mass loss estimates for Vincennes Bay", but is this accurate since you are showing ice shelf melt rates and calving rates? Might consider removing this qualifying sentence.

We have updated the figure caption for clarity, as follows:

*"Sub-ice shelf basal melt rates and ice-front migration for Vincennes Bay. (a) Long-term mean annual basal melt…"*

**Fig. 3** It is difficult to tell which experiments are bolded (I did not realize any were bold until I read the caption). Can you make this stick out more?

We have modified entries of basal melt and calving processing that are combined in hybrid experiments to improve readability of the figure.

**Fig. 6** Is panel-b showing the location of the grounding line along the flowline (e.g., in the Weertman $M_{Davidson}$ experiment, the grounding line was ~2.5 km retreated inland from year 20-25, but then advanced back to its present day position at year 30), or is this the retreat rate? It might be nice to briefly mention this in the figure caption or corresponding section of the paper.

Fig. 6b and Fig. 7d-f show the position of the grounding line along the flowline, relative to the position of the grounding line from the corresponding control experiment. We have updated the axis label to be "Δ Grounding line position (km)" and inverted the numbers to represent retreat as negative numbers and advance as positive numbers. We have updated the figure captions as follows:

*"Figure 6. … (b) Grounding line position calculated along the central flowline shown in Fig. 2c. Negative numbers represent grounding line retreat and positive numbers represent grounding line advance, relative to the Ctrl experiment. (c) …"*

*"Figure 7. … (d-f) Grounding line position calculated along the central flowline show in Fig. 2c. Negative numbers represent grounding line retreat and positive numbers represent grounding line advance, relative to the Ctrl experiment. (g-i) …"*

**Fig. 7** In the caption, I am a bit confused at your description of what the purple line is in panels d-f. Is the purple line the location of the 2020 grounding line along the flowline?

The purple line represents the position of the 2020 grounding line along the central flowline, relative to the position of the grounding line from the *Ctrl* experiments. Since the three different *Ctrl* experiments generate slightly different grounding line locations, we take the mean grounding line position from the three *Ctrl* experiments and calculate the 2020 grounding line position relative to this mean grounding line position. We have updated the figure caption as follows:

*"… Purple horizontal line in (d-f) shows the location of the 2020 grounding line (Picton et al., 2023), along the central flow line shown in Fig. 2c., relative to the mean grounding line location from all three basal friction Ctrl experiments."*

**Fig. 10** The orange line is a bit difficult to see given the colormap used. Can you try to make these stand out a bit more?

We have updated Fig. 10 to improve readability.

**References**

Chartrand, A. M. and Howat, I. M.: A Comparison of Contemporaneous Airborne Altimetry and Ice-Thickness Measurements of Antarctic Ice Shelves, Journal of Glaciology, pp. 1–14, https://doi.org/10.1017/jog.2023.49, 2023.

Davison, B. J., Hogg, A. E., Gourmelen, N., Jakob, L., Wuite, J., Nagler, T., Greene, C. A., Andreasen, J., and Engdahl, M. E.: Annual Mass Budget of Antarctic Ice Shelves from 1997 to 2021, Science Advances, 9, eadi0186, https://doi.org/10.1126/sciadv.adi0186, 2023.

Larour, E., Seroussi, H., Morlighem, M., and Rignot, E.: Continental Scale, High Order, High Spatial Resolution, Ice Sheet Modeling Using the Ice Sheet System Model (ISSM), Journal of

Geophysical Research: Earth Surface, 117, F01 022, https://doi.org/10.1029/2011JF002140, 2012.

McCormack, F. S., Cook, S., Goldberg, D. N., Nakayama, Y., Seroussi, H., Nias, I., An, L., Slater, D., and Hattermann, T.: The Case for a Framework for UnderStanding Ice-Ocean iNteractions (FUSION) in the Antarctic-Southern Ocean System, Elementa: Science of the Anthropocene, 12, 00 036, https://doi.org/10.1525/elementa.2024.00036, 2024.

Paolo, F. S., Gardner, A., Greene, C. A., and Schlegel, N.: MEaSUREs ITS_LIVE Antarctic Ice Shelf Height Change and Basal Melt Rates, Version 1, https://doi.org/10.5067/SE3XH9RXQWAM, 2022.

Picton, H. J., Stokes, C. R., Jamieson, S. S. R., Floricioiu, D., and Krieger, L.: Extensive and Anomalous Grounding Line Retreat at Vanderford Glacier, Vincennes Bay, Wilkes Land, East Antarctica, The Cryosphere, 17, 3593–3616, https://doi.org/10.5194/tc-17-3593-2023, 2023.

Shean, D. E., Joughin, I. R., Dutrieux, P., Smith, B. E., and Berthier, E.: Ice Shelf Basal Melt Rates from a High-Resolution Digital Elevation Model (DEM) Record for Pine Island Glacier, Antarctica, The Cryosphere, 13, 2633–2656, https://doi.org/10.5194/tc-13-2633-2019, 2019.

Zinck, A.-S. P., Wouters, B., Lambert, E., and Lhermitte, S.: Unveiling Spatial Variability within the Dotson Melt Channel through High-Resolution Basal Melt Rates from the Reference Elevation Model of Antarctica, The Cryosphere, 17, 3785–3801, https://doi.org/10.5194/tc-17-3785-2023, 2023.

---

## Referee Report (RR1)

**Comment on egusphere-2024-2060**

Benjamin Getraer

December 2024

**1 General Comments**

I greatly enjoyed reading this revised manuscript, and found the amendments made to the abstract, introduction, and discussion to greatly improve the clarity of the work. The considerable efforts which the authors made in these revisions, and in further experiments to answer questions raised in the initial reviews, add to the quality of the work and allow the significance of their results to shine more apparently to the reader.

I feel that the responses to the initial review are more than satisfactory, and feel much more confident in recommending this paper to be accepted for publication. In my re-reading of the paper, I found no major outstanding research questions which were unaddressed. However, I did find myself distracted by inconsistencies in formatting which I have flagged below. Most of these comments do not reflect on the scientific quality of the manuscript, and I do not think the manuscript needs further referee review.

**2 Specific Comments**

The following comments are minor spots which caught my attention as places where the manuscript could be improved.

1. The sentence starting on line 64 and ending in "quantifying the relative contributions of these processes on grounding line retreat and mass loss is essential" could be stronger and less vague if it specified explicitly what this knowledge is essential to. This comment is a question of motive and subjective writing style, and does not require correction.

2. The text does not always clearly distinguish between the "Vincennes Bay ice shelves" (i.e. line 48) and the "Vanderford Glacier ice shelf" (i.e. line 410). Figure 1 shows a region of connected floating ice labeled as the "Vincennes Bay ice shelf" (singular) with no specific label for the "Vanderford Glacier ice shelf." Elsewhere, the term "Vincennes Bay ice shelves" (plural) is used, and it is not clear whether this term is used to refer to all of the ice shelves in the study domain. In some places, it is implied that the plural "Vincennes Bay ice shelves" does not include the Underwood Glacier ice shelf (line 353). The language used does not prevent understanding the broader results and implications, but was somewhat confusing when trying to really read in depth. Minor edits to clarify this language and make it consistent with the labeling in Fig. 1 would be helpful.

3. Lines 69–71 use a numbered itemization to present the research questions. These could be dropped, which I think would improve the flow of the sentences without sacrificing clarity. Additionally, on line 73, "we are able to infer" can be replaced with simply "we infer," which is more direct. These comments are a question of subjective writing style, and do not require correction.

4. Figure 3 contains the word "bold" in bold font, which I found unnecessary and slightly distracting. Consider formatting in regular font.

5. Some figures, specifically Fig. 5 and Fig. 9, have labels with a much smaller font size that is difficult to read without zooming in. While I did not find this to be a significant obstacle, it does negatively affect the presentation quality of the paper.

**3 Technical Corrections**

The following corrections affect the presentation quality of the manuscript, which I think would benefit greatly from attention to these details. For technical corrections, I have followed the style set forth at `https://www.the-cryosphere.net/submission.html`. In instances where style recommendations are not clear, I have flagged issues of inconsistency in style within the manuscript.

1. **Volume above floatation in gigatonnes**
   The clarification of units in the figure captions, while helpful, does not resolve the fact that volume cannot be measured in units of mass. While I agree that gigatonnes are a helpful unit to use for the purposes of comparison with other studies, the measure describes mass above floatation, not volume. All of the analysis appears to use mass above floatation, and it is incorrect and unnecessary to call it a volume. The IPCC reports, when using units of gigatonnes, refer to changes in mass.

2. **Inconsistently hyphenated words**
   Compound words such as "ice shelf," "ice front," "ice sheet," and so on, are inconsistently hyphenated throughout the paper. As open and closed constructions are both valid, and TC does not appear to provide a clear style guide, the authors should choose one and stick with it for all instances of the word.

   There should also be consistency in the way attributive constructions of compound words are written. For example, if "sea level" becomes "sea-level" when used as a modifier (i.e. "sea-level rise," line 29), then all instances of open compound words should be hyphenated when used as modifiers. Examples where modifiers are not hyphenated are "grounding line retreat" (i.e. line 2), "ice front positions" (Fig. 4 caption, line 8).

   In addition, there are inconsistencies when compound words have hyphenated modifiers. The best construction makes it clear that the modifier applies to the entire compound word. For example, "sub-ice shelf" (i.e. line 4) is not the most clear construction. An open compound of "ice shelf" should become "sub–ice shelf" with an en dash. A closed compound of "ice-shelf" should become "sub-ice-shelf" with two hyphens.

   There are too many cases of these words to reference all of them here, but these issues occur in the main text, captions, figures, and abstract, and should be consistent. At the very least, there should be consistency for a given compound word in how it is written when it is a standalone noun, when it is the modifier, and when it is modified.

3. **Hyphens as ranges**
   Per the TC style guide, un-spaced en dashes should be used to denote ranges, but only in cases where no confusion with "$a$ minus $b$" is possible. A range of numbers should be specified as "$a$ to $b$" or "$a \ldots b$."
   For example, on line 190:

   > 10 - 100 m yr$^{-1}$

   should instead read:

   > 10 to 100 m yr$^{-1}$

   And on line 48:

   > Fig. 2a-b

   should instead read:

   > Fig. 2a–b

   These ranges appear throughout the main text, captions, and tables, and should be amended.

4. **Inconsistent capitalization in figure headings**
   Some figure and table headings and labels are in title case, while others are in sentence case. In some figures, these are inconsistent across similar labels and headings. These should be

consistent, and in sentence-style capitalization (i.e. capitalize the first word and proper nouns only). For example, in Fig. 1, "Ice Surface Velocity" and "Vincennes Bay Ice Shelf" are in title case, while "TG basin" and "VB basin" are in the correct sentence case. This issue occurs in Figs. 1–8, 10, and Table 1.

Additionally, "$x$" and "$y$" are coordinate system variables, and should be treated as mathematical characters when used as axis labels, not as text characters which would be capitalized.

**Small typos**

**Eq. 4** Missing space in "if $z_b \geq 0$"

**line 12** Spaced en dashes should be used for syntactic constructions. "...grounding line – more than twice current estimates – are needed...," instead of "...grounding line - more than twice current estimates - are needed...."

**lines 30 and 514** The word "glaciers" in "Totten and Vanderford Glaciers' flow configurations" should not be capitalized, as the plural no longer refers to a unique entity.

**lines 36 and 101** The phrases "marine ice sheet instability" and "shelfy-stream approximation" are not a proper nouns and should not be capitalized, even though their abbreviations are.

**lines 48, 189, 355, Fig. 4 caption lines 2 and 3** The character for a negative number is not the same as a text hyphen. For example, on line 48, "-9 to 34 m yr$^{-1}$" should read "$-9$ to 34 m yr$^{-1}$." I tried to find all instances of this issue, but it may occur elsewhere as well.

**line 71** The word "observed" seems to be missing the noun it is meant to modify, and could be replaced with "observed retreat" or simply "observations."

**line 121** In general, mathematical symbols such as $n$ are typeset in italics.

**line 150** Hyphen in "freely evolve."

**Fig. 2, lines 3 and 4** The words "green" and "red" should not be capitalized.

**Fig. 3** The experiment names are typeset in roman, but in other figures and main text are typeset in italics.

**line 169** Consider using "e.g." here, as the list presents representative examples rather than a clarification or restatement.

**line 214** Typesetting letters using the default math mode as opposed to italics can sometimes produce incorrect spacing: $VAF$ versus $VAF$. This spacing issue is reproduced elsewhere as well.

**line 255** Figure reference uses semicolon ("Fig. 5f; Fig. 7d") which is inconsistent with the form of the other figure references used (i.e. "Fig. 5f and Fig. 7d").

**Fig. 6 and 7, lines 2 and 4** "Grounding line flux," "grounding line retreat", and "ice volume" should not be capitalized where they do not begin a sentence.

**Fig. 6 and 7, line 4** SI and SI-accepted units (i.e. "gigatonne") must be written out when they are not accompanied by numbers.

**Fig. 7** Figure caption is overlapping the page number.

**line 353** The construction "Vincennes Bay / Underwood ice shelves" should either be written as "Vincennes Bay and Underwood ice shelves," or, if appropriate, to simply "Vincennes Bay ice shelves."

**line 358** Everywhere else in the paper "Vanderford Glacier ice shelf" is used as opposed to "Vanderford ice shelf."

**line 364** Here, "e.g." is followed by a comma. This is the familiar usage to me in American English, but "i.e." and "e.g." are used without commas everywhere else in the paper.

**line 389** To be consistent with the typesetting of other ranges in the paper, the units should appear once, after the second number (i.e. "25–30 %", not "25 % - 30 %").

**line 428** The clause containing "retreat from reduced buttressing forces may be observed at Vanderford Glacier over coming decades" does not need commas.

**line 429** Should read "Antarctic Surface Water," not "Antarctica Surface Water."

**line 453** Extra space before semicolon, after "calving laws."

**line 454** Missing coma after "perturbation period."

---

## Author Response (AR2)

**Assessing the sensitivity of the Vanderford Glacier, East Antarctica, to basal melt and calving**

**Author Responses to Referee Comments**

Lawrence A. Bird    Felicity S. McCormack    Johanna Beckmann

Richard S. Jones    Andrew N. Mackintosh

Dear Cheng Gong:

We thank both reviewers for their additional review of our manuscript and are pleased that they support its publication in The Cryosphere. Below, we provide a brief response (in blue) to the reviewers final comments (in black).

Sincerely,

Lawrence Bird and co-authors

**Reviewer #1 - Benjamin Gatraer**

**General comments**

I greatly enjoyed reading this revised manuscript, and found the amendments made to the abstract, introduction, and discussion to greatly improve the clarity of the work. The considerable efforts which the authors made in these revisions, and in further experiments to answer questions raised in the initial reviews, add to the quality of the work and allow the significance of their results to shine more apparently to the reader.

I feel that the responses to the initial review are more than satisfactory, and feel much more confident in recommending this paper to be accepted for publication. In my re-reading of the paper, I found no major outstanding research questions which were unaddressed. However, I did find myself distracted by inconsistencies in formatting which I have flagged below. Most of these comments do not reflect on the scientific quality of the manuscript, and I do not think the manuscript needs further referee review.

We are glad that the reviewer feels the revised manuscript is improved from the initial submission and is now ready for publication.

**Specific comments**

1. The sentence starting on line 64 and ending in "quantifying the relative contributions of
30 these processes on grounding line retreat and mass loss is essential" could be stronger and less vague if it specified explicitly what this knowledge is essential to. This comment is a question of motive and subjective writing style, and does not require correction.

We have amended this sentence to end with an explicit statement of why this work is important:

35 *"… quantifying the relative contributions of these processes on grounding line retreat and mass loss is essential to better understanding the vulnerability of the region to a changing climate."*

2. The text does not always clearly distinguish between the "Vincennes Bay ice shelves" (i.e. line 48) and the "Vanderford Glacier ice shelf" (i.e. line 410). Figure 1 shows a region of connected floating ice labeled as the "Vincennes Bay ice shelf" (singular) with
40 no specific label for the "Vanderford Glacier ice shelf." Elsewhere, the term "Vincennes Bay ice shelves" (plural) is used, and it is not clear whether this term is used to refer to all of the ice shelves in the study domain. In some places, it is implied that the plural "Vincennes Bay ice shelves" does not include the Underwood Glacier ice shelf (line 353). The language used does not prevent understanding the broader results and implications,
45 but was somewhat confusing when trying to really read in depth. Minor edits to clarify this language and make it consistent with the labeling in Fig. 1 would be helpful.

We have amended the use of "Vincennes Bay ice shelves" to "ice shelves within Vincennes Bay" or "ice shelves across Vincennes Bay" for clarity. We have added a label for the "Vanderford Glacier ice shelf" to Fig. 1 and amended to figure caption to include:

50 *The Vanderford Glacier ice shelf is not explicitly defined by Mouginot et al. (2017), but here we use "Vanderford Glacier ice shelf" to refer to the portion of the Vincennes Bay ice shelf that includes the main trunk of the Vanderford Glacier.*

3. Lines 69–71 use a numbered itemization to present the research questions. These could be dropped, which I think would improve the flow of the sentences without sacrificing
55 clarity. Additionally, on line 73, "we are able to infer" can be replaced with simply "we infer," which is more direct. These comments are a question of subjective writing style, and do not require correction.

We chose to leave the numbered itemisation of the research questions for clarity. We replaced the *"…are able to infer…"* on Line 73 with *"…can infer…"* to improve readability.

60    4. Figure 3 contains the word "bold" in bold font, which I found unnecessary and slightly distracting. Consider formatting in regular font.

We have reformatted the world "bold" to use regular font, as suggested by the reviewer.

  5. Some figures, specifically Fig. 5 and Fig. 9, have labels with a much smaller font size that is difficult to read without zooming in. While I did not find this to be a significant
65      obstacle, it does negatively affect the presentation quality of the paper.

We have increased the size of Fig. 5 and the size of axis labels on Fig 5. and Fig. 9 to improve readability of both figures.

**Technical corrections**

  1. **Volume above floatation in gigatonnes** The clarification of units in the figure cap-
70      tions, while helpful, does not resolve the fact that volume cannot be measured in units of mass. While I agree that gigatonnes are a helpful unit to use for the purposes of comparison with other studies, the measure describes mass above floatation, not volume. All of the analysis appears to use mass above floatation, and it is incorrect and unnecessary to call it a volume. The IPCC reports, when using units of gigatonnes, refer to changes
75      in mass.

We appreciate the reviewer highlighting this point again. To prevent ambiguity and improve clarity, we have decided to simply report all references to *VAF* and ice volume in $km^3$, rather than gigatonnes. We have updated Fig. 6 and 7, accordingly, and adjusted the values and units reported on line 237.

80    2. **Inconsistently hyphenated words** Compound words such as "ice shelf," "ice front," "ice sheet," and so on, are inconsistently hyphenated throughout the paper. As open and closed constructions are both valid, and TC does not appear to provide a clear style guide, the authors should choose one and stick with it for all instances of the word. There should also be consistency in the way attributive constructions of compound words are written.
85      For example, if "sea level" becomes "sea-level" when used as a modifier (i.e. "sea-level rise," line 29), then all instances of open compound words should be hyphenated when used as modifiers. Examples where modifiers are not hyphenated are "grounding line retreat" (i.e. line 2), "ice front positions" (Fig. 4 caption, line 8). In addition, there are inconsistencies when compound words have hyphenated modifiers. The best construction
90      makes it clear that the modifier applies to the entire compound word. For example, "sub-ice shelf" (i.e. line 4) is not the most clear construction. An open compound of "ice shelf" should become "sub—ice shelf" with an en dash. A closed compound of "ice-shelf" should become "sub-ice-shelf" with two hyphens. There are too many cases of these words to reference all of them here, but these issues occur in the main text, captions, figures, and
95      abstract, and should be consistent. At the very least, there should be consistency for a

given compound word in how it is written when it is a standalone noun, when it is the modifier, and when it is modified.

We have removed all reference to "ice-shelf" and "ice-front" and use simply "ice shelf" and "ice front" throughout. We chose to leave the use of "sub-ice shelf" for consistency, rather than using a closed compound formulation for this phrase.

3. **Hyphens as ranges** Per the TC style guide, un-spaced en dashes should be used to denote ranges, but only in cases where no confusion with "a minus b" is possible. A range of numbers should be specified as "a to b" or "a. . . b."

For example, on line 190:

10 - 100 m yr$^{-1}$

should instead read:

10 to 100 m yr$^{-1}$

And on line 48:

Fig. 2a-b

should instead read:

Fig. 2a–b

These ranges appear throughout the main text, captions, and tables, and should be amended.

We have updated all ranges to include the word "to" when the range includes numbers and to use en dashes when referring to ranges of figures.

4. **Inconsistent capitalization in figure headings** Some figure and table headings and labels are in title case, while others are in sentence case. In some figures, these are inconsistent across similar labels and headings. These should be consistent, and in sentence-style capitalization (i.e. capitalize the first word and proper nouns only). For example, in Fig. 1, "Ice Surface Velocity" and "Vincennes Bay Ice Shelf" are in title case, while "TG basin" and "VB basin" are in the correct sentence case. This issue occurs in Figs. 1–8, 10, and Table 1. Additionally, "x" and "y" are coordinate system variables, and should be treated as mathematical characters when used as axis labels, not as text characters which would be capitalized.

We have updated figures and tables to use sentence case throughout, and use "Easting" and "Northing" in place of "X" and "Y" for figure coordinate labels.

**Small typos**

We thank the reviewer for their thorough review and flagging these typographical errors. We have reviewed each of these small typos and made corrections/amendments accordingly. We note that the Fig. 7 caption still overlaps the page number; however, since the caption is still legible and final production of the manuscript will address this, we have not worked to reformat this here.

**Reviewer #2 - Tyler Pelle**

**General comments**

This is a re-review of Bird et al. (2024) manuscript on modeling of the Vanderford Glacier system in East Antarctica. Thank you to the authors for taking the time and care to complete a series of adjustments to the paper. I think the responses are comprehensive and that you have been balanced and fair in your responses. In particular, it was helpful to see the additional model runs assessing the impact of the 500 year spin-up versus starting the forcing perturbation after the 2 year relaxation period; this is excellent support for the methodological choices. There is good detail in the additional text, both in the manuscript and supplement. Overall, I think this paper provides valuable insights into the forcing required to drive observed magnitudes of retreat of Vanderford Glacier, with important implications for future observational and modeling studies. Thus, I think this will make an excellent contribution to The-Cryosphere. I only have one minor comment, but this will be very easy to address and after, I am happy to see this work published.

We are glad that the reviewer feels the revised manuscript is improved from the initial submission and is now ready for publication.

**Specific comments**

Figure 6b and associated explanation: Thank you for your explanation regarding retreat in the Weertman $M_{Davidson}$ simulation. Given that this looks like a large grounding line response in the figure but is only due to localized ungrounding of a small ice pocket, it would be helpful to add the figure-2 you provided in the reviewer response document to the supplement and then add a sentence in the main text that points to this figure and explains what is actually happening there.

We thank the reviewer for this suggestion and have added this figure to the Supplementary Information (Fig. S2), adjusted all other supplementary figure numbers accordingly, and added the following text to line 225:

*"…The $M_{Davison}$ experiment with the Weertman friction law shows rapid grounding line retreat of ~2.5 km (Fig. 6b); however, this arises due to minimal ungrounding of a localised area and is not indicative of notable or widespread grounding line retreat. Ice re-grounds rapidly once the perturbation is removed (Fig. S2)…"*

**References**

Mouginot, J., Scheuchl, B., and Rignot, E.: MEaSUREs Antarctic Boundaries for IPY 2007-2009 from Satellite Radar, Version 2, Boulder, Colorado USA. NASA National Snow and Ice Data Center Distributed Active Archive Center., https://doi.org/10.5067/AXE4121732AD, 2017.